# DPP9 is a novel component of the N-end rule pathway targeting the tyrosine kinase Syk

Daniela Justa-Schuch[1], Maria Silva-Garcia[1†], Esther Pilla[1†‡], Michael Engelke[2], Markus Kilisch[1], Christof Lenz[3,4], Ulrike Möller[1], Fumihiko Nakamura[5,6], Henning Urlaub[3,4], Ruth Geiss-Friedlander[1*]

[1]Department of Molecular Biology, University Medical Center Goettingen, Goettingen, Germany; [2]Institute of Cellular and Molecular Immunology, University Medical Center Goettingen, Goettingen, Germany; [3]Department of Bioanalytics, Institute of Clinical Chemistry, University Medical Center Goettingen, Goettingen, Germany; [4]Bioanalytical Mass Spectrometry Group, Max Planck Institute for Biophysical Chemistry, Göttingen, Germany; [5]Hematology Division, Department of Medicine, Harvard Medical School, Boston, United States; [6]Brigham and Women's Hospital, Boston, United States

*For correspondence: rgeiss@ gwdg.de

†These authors contributed equally to this work

Present address: ‡MRC Laboratory of Molecular Biology, Cambridge, United States

Competing interests: The authors declare that no competing interests exist.

**Abstract** The aminopeptidase DPP9 removes dipeptides from N-termini of substrates having a proline or alanine in second position. Although linked to several pathways including cell survival and metabolism, the molecular mechanisms underlying these outcomes are poorly understood. We identified a novel interaction of DPP9 with Filamin A, which recruits DPP9 to Syk, a central kinase in B-cell signalling. Syk signalling can be terminated by degradation, requiring the ubiquitin E3 ligase Cbl. We show that DPP9 cleaves Syk to produce a neo N-terminus with serine in position 1. Pulse-chases combined with mutagenesis studies reveal that Ser1 strongly influences Syk stability. Furthermore, DPP9 silencing reduces Cbl interaction with Syk, suggesting that DPP9 processing is a prerequisite for Syk ubiquitination. Consistently, DPP9 inhibition stabilizes Syk, thereby modulating Syk signalling. Taken together, we demonstrate DPP9 as a negative regulator of Syk and conclude that DPP9 is a novel integral aminopeptidase of the N-end rule pathway.

## Introduction

Proteases of the DPPIV family are serine aminopeptidases that remove a dipeptide from the N-terminus of substrates having a Pro or an Ala residue in the second position ($NH_2$-$X_{aa}$Pro or $NH_2$-$X_{aa}$Ala) (*Yu et al., 2010*; *Waumans et al., 2015*). DPP8 and DPP9 are the only known intracellular members of this family, sharing 60% homology with a higher conservation (>90% identity) in their active site (*Zhang et al., 2013*). Not surprisingly, DPP8 and DPP9 are similar in their biochemical properties (*Geiss-Friedlander et al., 2009*; *Connolly et al., 2008*). However, DPP9, but not DPP8, is rate-limiting for the hydrolysis of proline-containing peptides in the cytoplasm, and plays a role in maturation of antigenic peptides for presentation on MHC class I alleles (*Geiss-Friedlander et al., 2009*). Knock-in mice expressing an inactive variant of DPP9 die 8–24 hr after birth, demonstrating its importance for neonatal survival which is not compensated by DPP8 (*Gall et al., 2013*).

DPP9 shows a broad tissue distribution (*Qi et al., 2003*; *Ajami et al., 2004*), localizes to the cytosol (*Ajami et al., 2004*), nucleus (*Justa-Schuch et al., 2014*), and to the leading edge of migrating cells (*Zhang et al., 2015*). DPP9 is linked to several pathways including Akt signalling (*Yao et al., 2011*; *Pilla et al., 2013*), activation of pro-inflammatory M1 macrophages (*Matheeussen et al.,*

**eLife digest** Proteins are made up of building blocks called amino acids bonded together to form chain-like molecules. Around twenty different amino acids are used to make proteins, and enzymes called proteases can recognize specific pairs of amino acids in proteins and cut the bonds between them. Dipeptidylpeptidase 9 (or DPP9 for short) is a protease that removes two amino acids from the end of a protein, just as long the second amino acid is one of two specific kinds (namely, an alanine or a proline). The DPP9 protease influences a range of processes in the cell including cell death, signaling and survival. Indeed, mice born with an inactive version of DPP9 die shortly after birth, but it is not known why this happens.

Justa-Schuch et al. investigated how the protease DPP9 controls processes inside cells and found an unexpected connection between DPP9 and another protein called Syk. The Syk protein is found in immune cells called B cells, and becomes highly activated whenever these cells are stimulated. Once activated Syk changes the activity of many proteins, affecting which genes are switched on and how the B cell moves and divides.

By using DPP9 as a kind of bait, Justa-Schuch et al. found human proteins that bind to the protease. This search identified a protein called Filamin A that interacted with DPP9, placing DPP9 close to Syk, which also binds to Filamin A. Further experiments showed that when DPP9 was located close to Syk, it cut the end of Syk. This cut left the Syk protein with a different amino acid exposed at its end, which in turn made it susceptible to being broken down inside the cell.

Justa-Schuch et al. went on to show that DPP9 preferentially cleaved the active form of Syk. Since cleaved Syk was subsequently broken down, DPP9 acts as a shut-off mechanism for Syk after the B cell has been stimulated. The findings show that DPP9 can influence how much and how long the B cell responds to stimulation. Inhibitors of DPP9 may therefore be useful for stabilizing Syk, which is known to stop specific tumors from growing. Future work will investigate the mechanisms that control how Filamin A, DPP9 and Syk interact.

2013), cell migration (*Zhang et al., 2015*) and apoptosis of specific cell lines (*Matheeussen et al., 2013*; *Spagnuolo et al., 2013*). The molecular mechanisms leading to these different outcomes are poorly understood.

For a profounder understanding of its molecular functions, we screened for interacting partners of DPP9, reasoning that these will include substrates and regulators of this peptidase. Previously, we identified SUMO1 as a novel DPP9 associating protein, and demonstrated that SUMO1 acts as an allosteric activator of DPP9 (*Pilla et al., 2013*, *2012*). Now we report filamin A (FLNA) as a novel interacting partner of DPP9, which we identified in a yeast two-hybrid assay.

FLNA is an actin binding protein that cross-links actin filaments into orthogonal networks, and is important for cell stiffening, cell adhesion and cell migration (*Nakamura et al., 2011*). An actin-binding domain is located at the amino terminus of FLNA, followed by 24 Ig-like repeats. Repeat 24 is involved in FLNA dimerization, whereas the other repeats are important for the rod-like structure of FLNA (*Figure 1A*) (*Nakamura et al., 2007*), and participate in numerous interactions with membrane, cytoplasmic and nuclear proteins (*Zhou et al., 2010*). Consequently, FLNA influences several cellular signalling events. Most proteins identified so far bind preferentially to FLNA repeats 16–24 (*Yue et al., 2013*). Surprisingly, we mapped the interaction between DPP9 and FLNA to repeats 5–7. Apart from DPP9, so far only the non-receptor tyrosine kinase Syk is known to bind to FLNA repeat 5 (*Falet et al., 2010*).

Although it is also expressed in non-haematopoietic cells, Syk is best characterized as an essential component in B cell receptor (BCR) mediated signalling. Syk is activated upon BCR engagement, and consequently initiates a cascade of events by phosphorylating its downstream effectors (*Mócsai et al., 2010*; *Geahlen, 2009*). The interaction between FLNA and Syk was identified in platelets, where loss of FLNA resulted in reduced signalling and mislocalization of Syk (*Falet et al., 2010*). Here we show that DPP9 forms a complex with FLNA and Syk. In this complex FLNA acts as a recruiting factor linking DPP9 to Syk, resulting in cleavage of the Syk N-terminus by DPP9, thus affecting Syk stability and Syk dependent signal transduction.

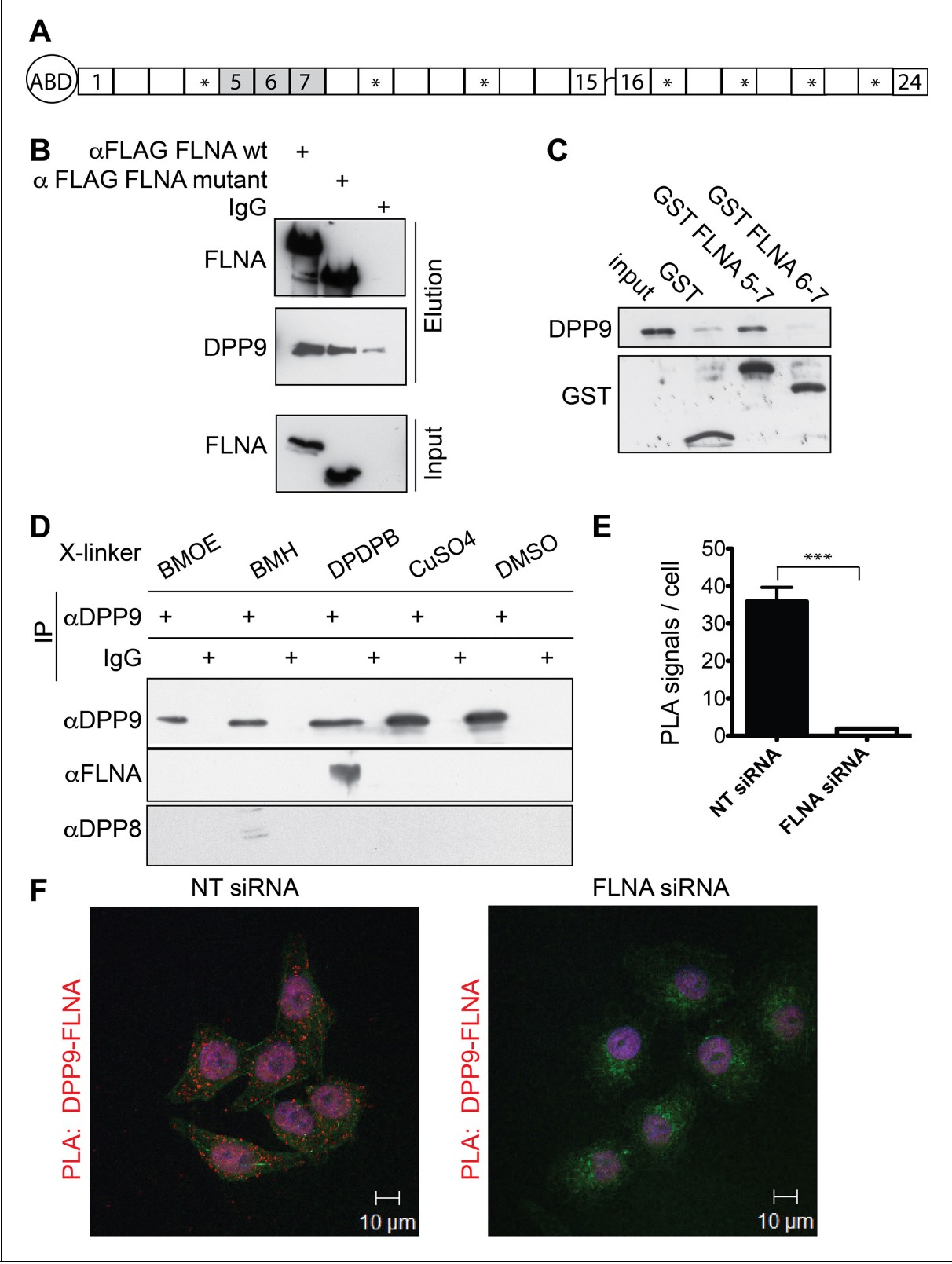

**Figure 1.** Filamin A (FLNA) is a novel DPP9 interacting protein. (**A**) Schematic representation of FLNA structure including numbering of the Ig-like domain repeats, and labelling of the actin-binding domain (ABD). The asterisks mark the repeats lacking in the FLNA variant form used in (**B**). (**B**) Pull-down assays showing direct interaction between recombinant DPP9 and recombinant FLAG tagged wt FLNA or a mutated form of FLAG-FLNA (lacking repeats 4, 9, 12, 17, 19, 21, and 23). Shown is a representative result of at least three independent experiments. (**C**) Recombinant DPP9 binds directly to

*Figure 1 continued on next page*

*Figure 1 continued*

GST- FLNA construct containing repeats 5–7 but not to GST-FLNA construct containing repeats 6–7. Shown is a representative result of at least three independent experiments. (D) Co-immunoprecipitation of endogenous FLNA with endogenous DPP9 from HeLa cells treated with different cross-linkers. Binding was observed in the presence of the sulfhydryl cross-linker DPDPB. Shown is a representative result of at least three independent experiments. To control for the specificity of the cross link, we blotted for DPP8, which did not bind to DPP9 in the presence of DPDPB (E) Quantification of the proximity ligation assay (in situ PLA) visualizing DPP9-FLNA interaction in HeLa cells treated with FLNA silencing oligos or non-targeting (NT) siRNAs for control shown in (F). The number of PLA signals per cell were quantified in a blinded manner using the Duolink ImageTool software (SIGMA). Data are represented as mean ± SEM. Signals of more than 130 cells were quantified for each condition respectively. Statistical analysis was carried out by an unpaired two-tailed t test (***p<0.0005). (F) PLA showing interaction of DPP9 with FLNA in HeLa cells. Each red dot represents a single FLNA-DPP9 interaction. The number of PLA signals is significantly decreased in cells silenced for FLNA compared to cells treated with NT siRNA. Actin filaments are stained in green, nuclei were visualized by using HOECHST. Shown are representative images of at least three independent PLA experiments.

The following figure supplement is available for figure 1:

**Figure supplement 1.** Co localization of DPP9 and FLNA.

## Results

### Filamin A – a novel DPP9 interacting protein

To identify novel proteins that interact with DPP9, a human placenta library was screened in a yeast two-hybrid assay with full-length DPP9 as bait. FLNA was one of the most promising candidates, which was identified in 12 of 142 processed clones. The binding surface was mapped to residues 748–907 of FLNA, corresponding to FLNA repeats 5–7. We verified direct interaction between FLNA and DPP9 by performing pull-down assays with the recombinant proteins (*Figure 1B*). Only background binding of DPP9 was observed in the absence of FLNA. In line with the yeast two-hybrid assay showing the importance of repeats 5–7, DPP9 bound to an FLNA deletion construct lacking repeats 4, 9, 12, 17, 19, 21, 23 (*Figure 1A and B* 'FLNA variant'), which were previously mapped for several other FLNA interactions (*Nakamura et al., 2011*; *Yue et al., 2013*). DPP9 also associated with a truncated version of FLNA expressing only repeats 5–7, but not with a shorter construct lacking repeat 5 (*Figure 1C*).

The interaction between DPP9 and FLNA was further analysed by performing co-immunoprecipitation assays against the endogenous proteins. To stabilize transient interactions, HeLa cells were incubated with various cross-linkers. Specific co-immunoprecipitation of FLNA with DPP9 was detected in cells treated with the sulfhydryl (-SH) cross-linker DPDPB, containing a spacer of 19.9 Å (*Figure 1D*). Co-immunofluorescence microscopy images taken from HeLa cells decorated with antibodies targeting DPP9 and FLNA showed an overlap in the cellular localization of these two proteins (*Figure 1—figure supplement 1A and B*). Next, in situ proximity ligation assays (PLA) were performed to visualize the association of endogenous DPP9 and FLNA in cells. Notably, we detected several distinct PLA dots in HeLa cells, each dot representing a single DPP9-FLNA interaction event. The number of PLA signals in control cells silenced for FLNA was strongly reduced compared to non-silenced cells (*Figure 1E and F*). Taken together, these results describe a novel and direct interaction between DPP9 and FLNA, which requires FLNA repeat 5 and is readily detected in cells.

### FLNA acts as a scaffold linking DPP9 to Syk - a novel DPP9 substrate

Apart from DPP9, only Syk is known so far to interact with FLNA repeat 5 (*Falet et al., 2010*). By PLA we show that the association between FLNA and Syk, which was first identified in platelets, is conserved in HeLa cells (*Figure 2—figure supplement 1*). Strikingly, clear PLA signals were also detected in HeLa cells decorated with antibodies against endogenous DPP9 and Syk, suggesting an interaction between these proteins in cells (*Figure 2A and B*).

Interestingly, the amino terminus of Syk contains a DPP9 consensus cleavage site Xaa-Pro/Ala (*Connolly et al., 2008*; *Ajami et al., 2004*) (*Figure 2—figure supplement 2A and B*) with an alanine in second position (Met-Ala), suggesting that Syk may be a DPP9 substrate. Therefore, we tested whether DPP9 interacts with the amino terminus of Syk by conducting Surface Plasmon Resonance (SPR) assays. Recombinant DPP9 was immobilized on a chip and probed for interaction with a

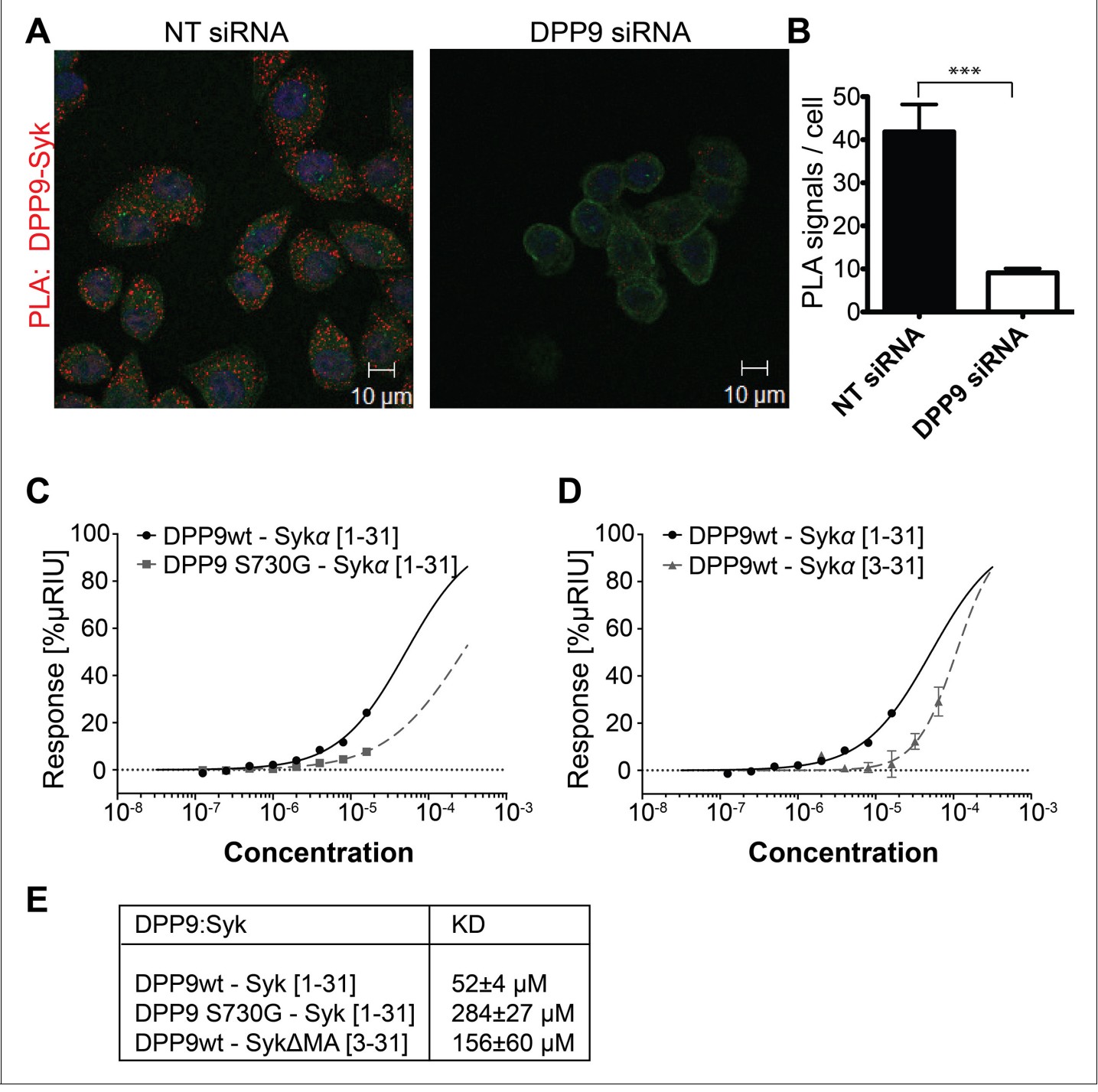

**Figure 2.** DPP9 interacts with the amino terminus of the tyrosine kinase Syk. (**A**) PLA in HeLa cells showing interaction of DPP9 with Syk. The number of PLA signals (red) representing DPP9-Syk interaction is significantly reduced in cells silenced for DPP9 compared to cells treated with NT siRNAs. Shown are representative images of at least three independent PLA experiments. Actin filaments are stained in green, nuclei are visualized with HOECHST staining. (**B**) Quantification of the PLA DPP9-Syk shown in (**A**). Data are represented as mean ± SEM. Signals of more than 150 cells for each condition were quantified respectively in a blinded manner using the Duolink ImageTool software (SIGMA). Statistical analysis was carried out by an unpaired two-tailed t test (***p<0.0005). (**C**) Surface Plasmon Resonance (SPR) assays showing direct interaction between DPP9 wild type and a synthetic peptide covering the first 31 amino acids of Syk (1–31). The binding affinity of the Syk (1–31) peptide is lower towards the inactive DPP9 variant (DPP9 S730G). Depicted are equilibrium binding isotherms obtained from at least three repetitions for respective interaction pairs of recombinant DPP9 and DPP9 S730G with Syk (1–31) peptides. DPP9 was immobilized at the chip surface and the Syk (1–31) peptide was injected over the surface with concentrations varying from 16 µM to 0.125 µM. Binding affinities were calculated using Graph Pad Prism 6.0. The Error is displayed as SEM. (**D**) Surface Plasmon

*Figure 2 continued on next page*

*Figure 2 continued*

Resonance (SPR) assays showing that the interaction of DPP9 with the peptide corresponding to Syk N-terminus requires the first two residues in Syk N-terminus. Depicted are equilibrium binding isotherms obtained from at least three repetitions for respective interaction pairs of recombinant DPP9 with Syk (1–31) or Syk (3–31) peptides. Recombinant His-tagged DPP9 was immobilized on the chip surface and the Syk (1–31) or Syk (3–31) peptide was injected over the surface with concentrations varying from 16 µM to 0.125 µM for Syk (1–31) and 32 µM to 1 µM for Syk (3–31). Binding affinities were calculated as described in (C). (E) Table summarizing the $K_D$ values.

The following figure supplements are available for figure 2:

**Figure supplement 1.** Interaction of FLNA with Syk is conserved in HeLa cells.

**Figure supplement 2.** DPP9 cleaves after a Xaa-Pro/Ala.

synthetic Syk (1–31) peptide covering the first 31 amino acids of full-length Syk (MA↓SSGMADSANH LPFFFGNITREEAEDYLVQ). Based on published solved structures, the amino terminus of Syk includes an unstructured region followed by an α-helix (residues 22–31) (*Grädler et al., 2013*; *Fütterer et al., 1998*). Of note, amino acids 1–7 are not resolved in any determined structure of Syk, suggesting that this region is flexible and thus accessible for interactions. We observed direct binding of DPP9 to the Syk (1–31) peptide with a $K_D$ of 52 ± 4 µM (*Figure 2C and E*), suggesting a dynamic and transient interaction. Furthermore, an inactive DPP9 variant in which the serine in the active site was mutated to a glycine residue (DPP9 S730G) showed more than a five-fold reduction in its affinity to the Syk (1–31) peptide ($K_D$ of 284 ± 27 µM) compared to the wild type protein (*Figure 2C and E*). A lower affinity ($K_D$ of 156 ± 60 µM) was also observed between wild type DPP9 and a Syk△MA (3–31) peptide, which lacks the first two amino acids Met-Ala (*Figure 2D and E*). In summary, these results show a direct interaction between DPP9 and Syk N-terminus, which involves the active site of DPP9, suggesting that Syk is a DPP9 substrate.

Next in vitro cleavage assays were performed to test for processing of the Syk N-terminus peptide (1–31) by recombinant DPP9. Mass spectrometry analysis of these reactions revealed that after six hours incubation, DPP9 had cleaved more than 99% of this peptide to remove the first two residues (*Figure 3A*). Processing was strongly reduced to only 6.2% in control samples treated with the allosteric DPP8/9 peptide inhibitor SLRFLYEG (*Pilla et al., 2013*). No cleavage was observed in the presence of the inactive variant DPP9 S730G (*Figure 3A*).

To further test whether DPP9 activity affects its interaction with Syk, HeLa cells were treated with SLRFLYEG. Previously we demonstrated that this inhibitor can be delivered into cells if it is pre-incubated with cell penetrating peptides (Pep1) to form a non-covalent Pep-1-SLRFLYEG complex. Once in cells this complex dissociates leading to inhibition of DPP9 by SLRFLYEG (*Pilla et al., 2013*). Consistently, exposure of cells to SLRFLYEG resulted in a significant reduction in PLA signals corresponding to DPP9-Syk interaction events, compared to the control cells treated with the carrier peptide only (*Figure 3B and C*). Likewise, treatment of cells with the competitive DPP9 inhibitor 1G244 (*Wu et al., 2009*) also led to a clear decrease in the number of Syk-DPP9 PLA signals (*Figures 3D and E*, *Figure 3—figure supplement 1*). Of note 1G244 and all other available DPP9 inhibitors also target DPP8 due to the high conservation in the active site of both enzymes (*Van Goethem et al., 2011*). For control, we measured the association of DPP9 with FLNA, which was not significantly altered by the 1G244 treatment (*Figure 3F and G*). These results demonstrate that Syk, but not FLNA, requires access to the active site of DPP9 for interaction. Taken together, we conclude that Syk is a novel DPP9 substrate.

What is the role of FLNA for the DPP9-Syk interaction? Strikingly, immunofluorescence microscopy images show a drastic change in the cellular localization of DPP9 in FLNA silenced cells compared to control cells treated with non-targeting siRNA (*Figure 4A and B*). In particular, upon FLNA silencing, DPP9 was no longer observed at the plasma membrane and was detected less in the cytosol, showing elevated levels in the nucleus (*Figure 4A and B*). These results demonstrate that FLNA has a strong impact on the cellular localization of DPP9. We further tested the importance of FLNA for the DPP9-Syk interaction by performing PLAs of DPP9-Syk in FLNA silenced cells, and compared to cells treated with non-targeting siRNA. We detected a significant reduction in the number of DPP9-Syk interaction events in the FLNA silenced cells as indicated by a lower number of PLA dots

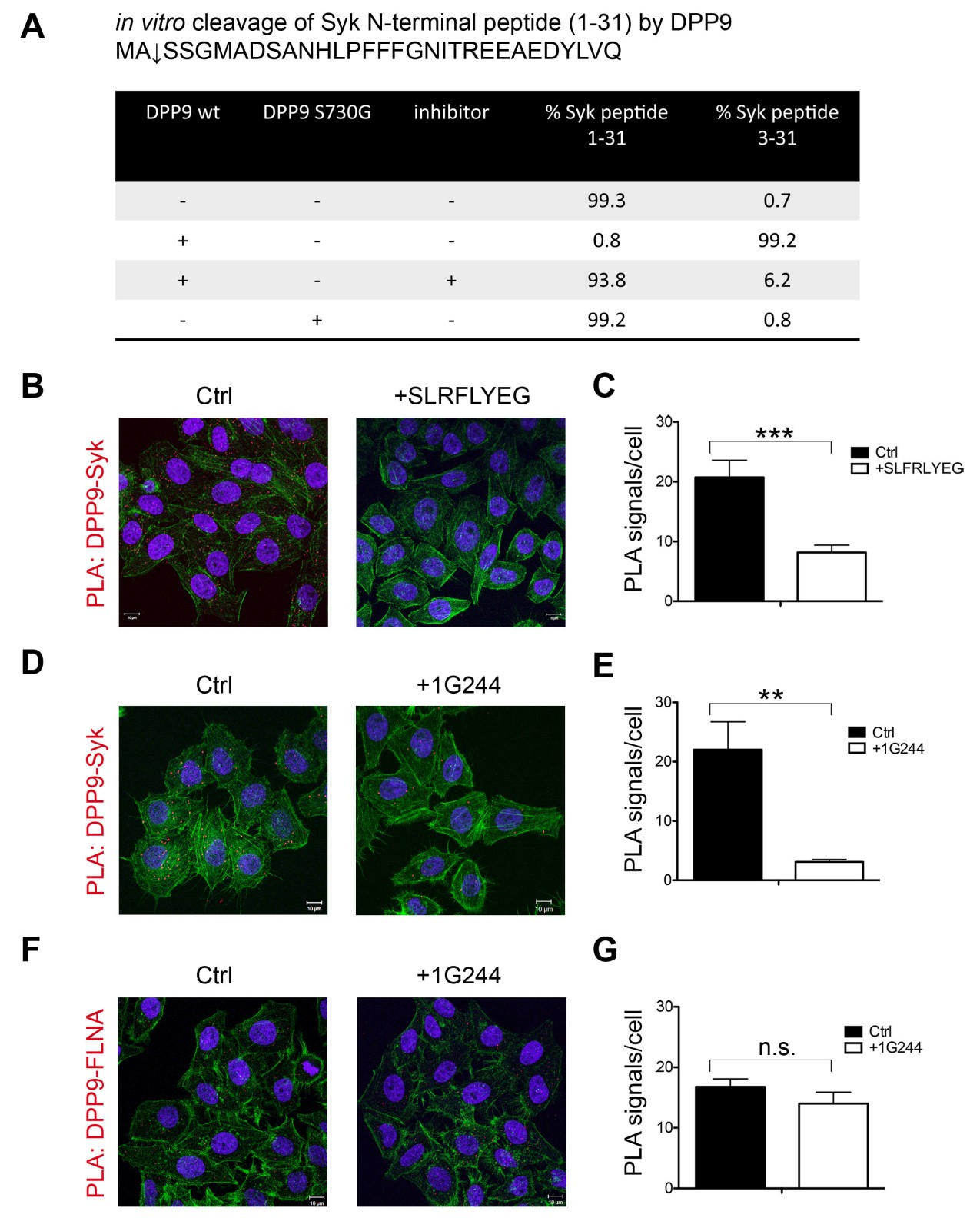

**A** *in vitro* cleavage of Syk N-terminal peptide (1-31) by DPP9
MA↓SSGMADSANHLPFFFGNITREEAEDYLVQ

| DPP9 wt | DPP9 S730G | inhibitor | % Syk peptide 1-31 | % Syk peptide 3-31 |
|---|---|---|---|---|
| - | - | - | 99.3 | 0.7 |
| + | - | - | 0.8 | 99.2 |
| + | - | + | 93.8 | 6.2 |
| - | + | - | 99.2 | 0.8 |

**Figure 3.** Syk is a novel substrate of DPP9. (**A**) In vitro cleavage of a synthetic Syk peptide corresponding to the N-terminus of Syk (1–31) by recombinant DPP9. 50 μM of a synthetic Syk (1–31) peptide was incubated for 6 hr, either alone or with 130 nM DPP9. For control 10 μM allosteric DPP9 inhibitor SLRFLYEG was added in addition to 130 nM DPP9 and (6 hr). An additional control included the peptide and the inactive DPP9 S730G variant. Samples were analysed by high resolution liquid chromatography/tandem mass spectrometry in triplicate. Quantitation was achieved by extracting ion

*Figure 3 continued on next page*

*Figure 3 continued*

chromatograms and integrating peak areas for the most abundant 3+ charge state of the intact 1–31 ([M+3H]$^{3+}$ *m/z* 1149.8589) and the cleaved 3–31 ([M+3H]$^{3+}$ *m/z* 1082.4997) peptides. The identities and retention times of the peptides were established by accurate mass measurement and product ion spectra (data not shown). (**B–G**) PLA assays showing that the interaction between DPP9 and Syk requires the active site of DPP9. Shown are representative images with the corresponding quantifications of at least three independent PLA experiments. Actin filaments are stained in green, and nuclei were visualized by using HOECHST. The number of PLA signals (red dots) per cell were quantified in a blinded manner using the Duolink ImageTool software (SIGMA). Signals of more than 300 cells were quantified for each condition respectively. Statistical analysis was carried out by an unpaired two-tailed t test (**p<0.005; ***p<0.0005; n.s = not significant). (**B**) The interaction between DPP9 and Syk is markedly decreased in HeLa cells treated with 10 µM SLRFLYEG compared to control cells treated with DMSO. (**C**) Quantification of the PLA DPP9-Syk shown in (**B**). Data are represented as mean ± SEM. (**D**) The number of PLA signals representing DPP9-Syk interactions per cell is reduced upon treatment of HeLa cells with the competitive DPP8/9 inhibitor 1G244 (10 µM, for 5 min) compared to control cells treated with DMSO. (**E**) Quantification of the PLA DPP9-Syk shown in (**D**). Data are represented as mean ± SEM. (**F**) The interaction of DPP9 with FLNA is not significantly altered upon treatment of HeLa cells with 1G244 (10 µM, 30 min) compared to control cells treated with DMSO. (**G**) Quantification of the PLA DPP9- FLNA shown in (**F**). Data are represented as mean ± SEM.

The following figure supplement is available for figure 3:

**Figure supplement 1.** Inhibition of DPP activity in HeLa cells with 1G244.

(*Figure 4C and D*). Taken together, we conclude that the presence of FLNA is important for the ability of DPP9 and Syk to interact in cells, which eventually leads to Syk processing.

## DPP9 determines Syk stability by exposing an amino terminal serine

To study the outcome of Syk cleavage by DPP9 we turned to a stable HeLa cell line for DPP9 silencing (DPP9-kd). This cell line shows a 40% reduction in its capacity to cleave the artificial substrate Gly-Pro-AMC (GP-AMC), which can be cleaved by DPP8, DPP9 and DPPIV. Additionally, we detected at least a 50% reduction in DPP9 protein levels in these cells compared to the control parental cell line (*Figure 5A and B*). Surprisingly, although we transfected wild type and DPP9-kd cells with equal amounts of plasmid DNA encoding C-terminal FLAG tagged Syk (Syk-FLAG), higher Syk protein levels were observed in the DPP9-kd cell line (*Figure 5B*). These unequal protein levels of the transiently transfected Syk constructs suggested that Syk may be more stable in the DPP9-kd cell line than in the wild type parental cell line. To test this hypothesis, we performed cycloheximide (CHX) chase experiments of DPP9-kd and the parental cells transfected with the Syk-FLAG construct. Cells were treated with CHX to inhibit protein synthesis ('pulse', time 0), and samples were taken at different time points for the chase. We found that Syk-FLAG is degraded in HeLa cells with a half-life of 6 hr. Remarkably, Syk-FLAG was more stable in the DPP9-kd cells compared to the control parental HeLa cells (*Figures 5C and D*). These results suggest that processing of Syk by DPP9 and exposure of a neo Syk N-terminus with serine in position 1 (MA↓S) leads to Syk degradation, possibly by the N-end rule pathway.

The N-end rule pathway degrades proteins based on their first N-terminal residue (N-degron) (*Varshavsky, 2011*). N-end rule substrates are ubiquitinated by specific E3 ubiquitin ligases (N-Recognins) that recognize the N-degrons in proteins leading to their proteasome degradation (*Tasaki et al., 2005*, *2009*; *Hwang et al., 2010a*). To test whether the serine that is exposed upon DPP9 cleavage indeed determines Syk stability, we mutated this residue to a glycine (SykS3G) or a valine (SykS3V). Following processing by DPP9 these Syk constructs would differ only in the first amino acid in their N–terminus, which would be either Ser in wild type Syk, Gly or Val in the variant proteins. We chose these residues since substrates with Ser, Gly or Val in the P1' position (after the cleavage site) are processed by DPP9 with a similar efficiency (*Geiss-Friedlander et al., 2009*). Additionally, we mutated the alanine in position 2 to an aspartic acid (SykA2D). This mutation destroys the DPP9 cleavage site, and thus should prevent Syk processing by DPP9 (*Connolly et al., 2008*). The different Syk constructs were transfected into HeLa cells and their stability was compared by CHX chase experiments. Strikingly, the single mutation of Ser 3 to a Gly or a Val clearly increased the half-life of the SykS3G and SykS3V variants (*Figure 5E and F*), compared to the wild-type protein. Furthermore, an increased half-life of Syk was also measured for the SykA2D variant, which cannot be processed by DPP9 (*Figure 5G and H*). These results highlight the importance of Syk

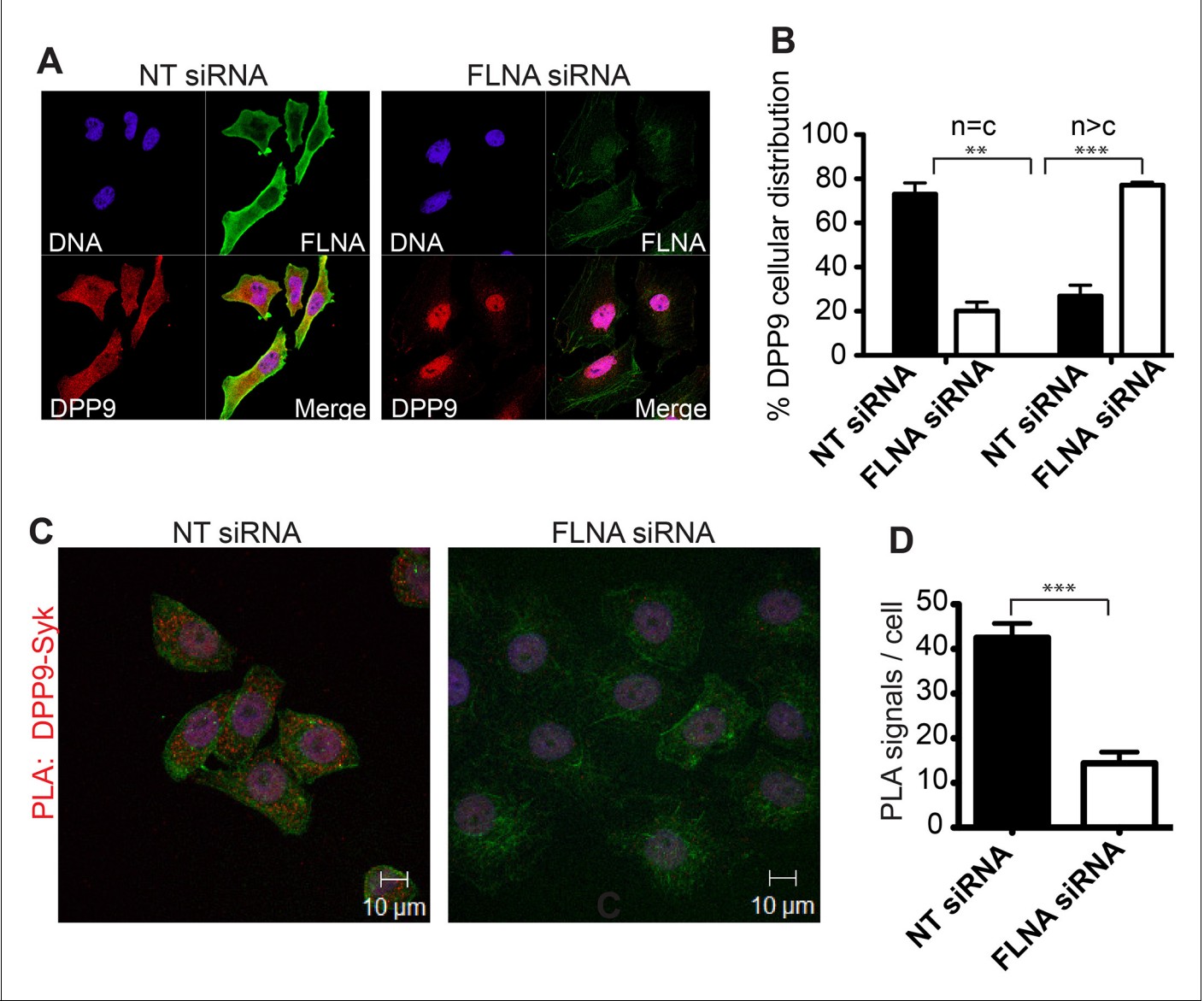

**Figure 4.** FLNA - a scaffold linking DPP9 to Syk. (A) Immunofluorescence images of HeLa cells showing that the cellular localization of DPP9 is altered in FLNA silenced cells. DPP9 is shown in red, FLNA in green and nuclei are visualized with HOECHST staining (blue). Shown are representative images of at least three independent experiments. (B) Quantification of the immunofluorescence shown in (A). n = c: DPP9 is equally distributed in the cytosol and nucleus; n > c stronger DPP9 staining in the nucleus compared to the cytosol. Data are represented as mean ± SEM. Signals of more than 100 cells for each condition were quantified respectively. Statistical analysis was carried out by an unpaired two-tailed t test (\*\*p<0.005; \*\*\*p<0.0005). (C) The interaction of DPP9 with Syk depends on the presence of FLNA in HeLa cells. The number of PLA signals (red dots) per cell is markedly decreased in cells treated with FLNA silencing oligonucleotides compared to control cells treated with NT siRNA oligonucleotides. Shown are representative images of at least three independent PLA experiments. (D) Quantification of the PLA DPP9-Syk shown in (C). Data are represented as mean ± SEM. The number of PLA signals (red dots) per cell were quantified in a blinded manner using the Duolink ImageTool software (SIGMA). Signals of more than 115 cells were quantified for each condition respectively. Actin filaments are stained in green, nuclei were visualized with HOECHST staining. Statistical analysis was carried out by an unpaired two-tailed t test (\*\*\*p<0.0005).

cleavage by DPP9 and the destabilizing effect of the serine that is exposed in the N-terminus of Syk upon cleavage.

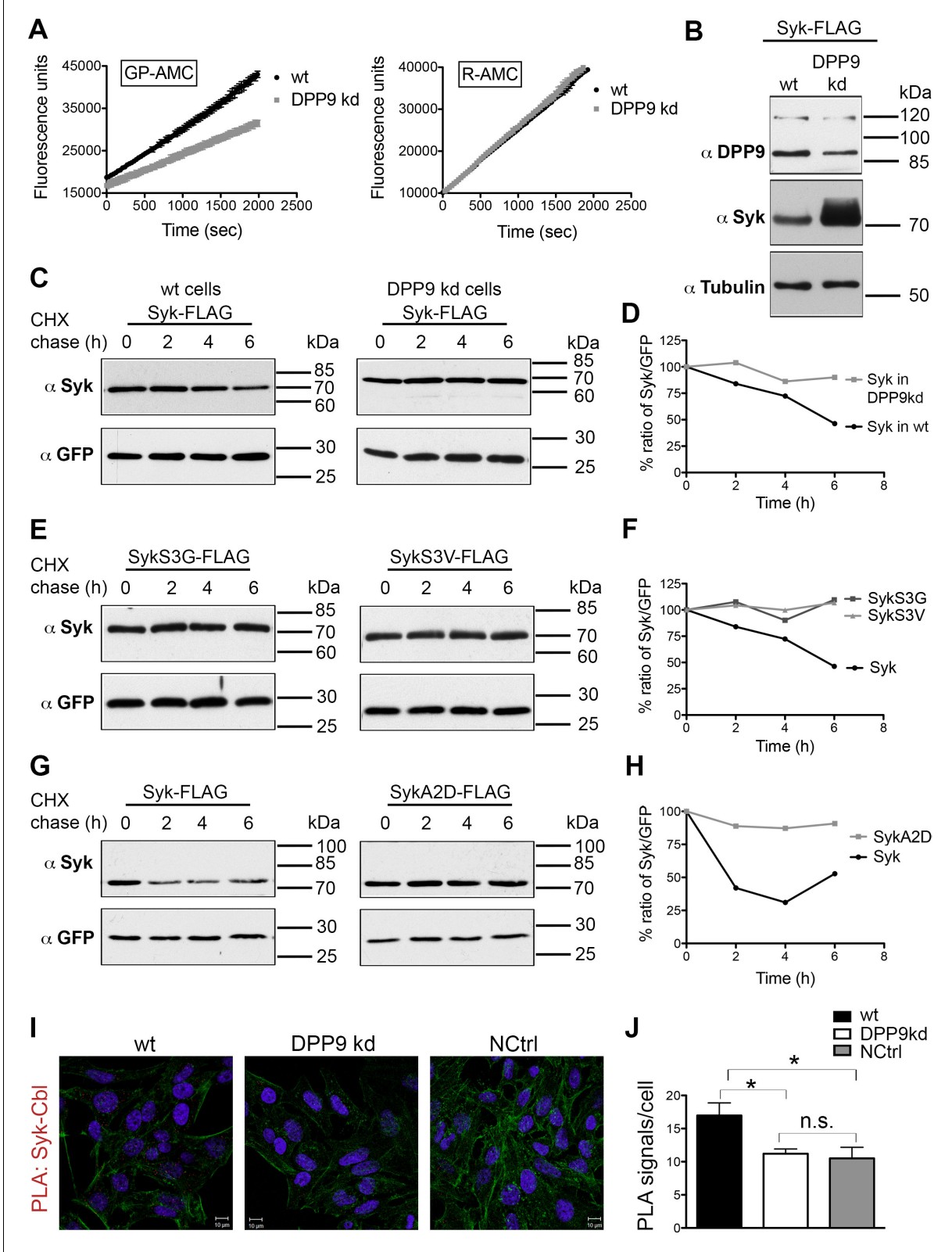

**Figure 5.** DPP9 determines Syk stability by exposing an N-terminal serine. (**A**) Reduced DPP activity in cytosolic extracts of HeLa cells with a stable silencing of DPP9 (DPP9-kd). Cell extracts either from HeLa wt or DPP9-kd cells (5 μg) were tested for DPP activity using the artificial DPP substrate GP-AMC (250 μM) or the unrelated substrate R-AMC (50 μM). Fluorescence was measured over time. The experiments were performed at least three times, each time in triplicates. Shown is a representative, data are represented as mean ± SEM. (**B**) Higher steady-state levels of Syk in DPP9-kd compared to

*Figure 5 continued on next page*

Figure 5 continued

HeLa wt cells. Western blot analysis of cell lysates from HeLa wt or DPP9-kd cells transfected with 1 µg C-terminally FLAG-tagged full-length Syk. Shown is a representative result of at least three independent experiments. Tubulin was analysed as a loading control. (C) Syk stability is determined by DPP9. HeLa wt or DPP9-kd cells were transfected with C-terminally FLAG-tagged full-length Syk and subjected to Cycloheximide (CHX) chase assays. GFP was analysed as a transfection and loading control. Shown is one representative result of at least three independent experiments. (D) Quantification of the Western blot results shown in (C). The ratio of Syk/GFP at time 0 hr was normalized to 100%. For signal quantification GelQuant. NET software provided by biochemlabsolutions.com was used. (E) Syk stability is determined by the serine that is exposed after cleavage by DPP9. HeLa wt cells were transfected with different Syk constructs: Syk-FLAG, SykS3G-FLAG or SykS3V-FLAG and subjected to CHX chase assays. Shown is one representative result of at least three independent experiments. (F) Quantification of the Western blot results shown in (E), as described in (D). (G) Syk stability is increased by mutating the A at position 2 in the DPP9 cleavage site to a D (SykA2D). HeLa wt cells transfected with Syk-FLAG or SykA2D-FLAG were subjected to CHX chase assays. Shown is one representative result of at least three independent experiments. (H) Quantification of the Western blot results shown in (G), as described in (D). (I) Reduced interaction events between Syk and Cbl in the absence of DPP9. Interaction of Syk with Cbl was visualized by PLA in HeLa wt and DPP9-kd cells. The number of PLA signals representing Syk-Cbl interactions per cell was reduced in DPP9-kd cells. For control, cells were treated with only one primary antibody (anti Syk). (J) Quantification of the PLA Syk-Cbl in HeLa cells shown in (I). Data are represented as mean ± SEM. The number of PLA signals per cell were quantified in a blinded manner using the Duolink ImageTool software (SIGMA). Signals of more than 100 cells were quantified for each condition respectively using the Duolink ImageTool (SIGMA). Statistical analysis was carried out by an unpaired two-tailed t test (*$p < 0.05$; n.s = not significant).

## DPP9 leads to Syk degradation in B cells upon BCR-engagement

In B cells Syk signalling is initiated by the binding of an antigen to the BCR. This engagement of the BCR leads to receptor aggregation and phosphorylation of tyrosine residues in transmembrane ITAM containing proteins (immunoreceptor tyrosine based activation motifs). Syk is recruited to phosphorylated ITAMS via two SH2 domains in its amino terminus, resulting in Syk phosphorylation and activation (*Mócsai et al., 2010*). Syk is negatively regulated by the E3 ubiquitin ligase Cbl (*Ota et al., 2000*; *Paolini et al., 2001*; *Joazeiro et al., 1999*; *Rao et al., 2001a*; *Sohn et al., 2003*). Proteasome inhibition in natural killer cells results in the accumulation of ubiquitinated Syk species, suggesting that ubiquitination targets the kinase for proteasomal degradation (*Paolini et al., 2001*). Performing PLAs we established an interaction between endogenous Cbl and Syk in HeLa cells (*Figure 5I*). Strikingly, the number of PLA signals corresponding to a Syk-Cbl interaction were significantly lower in the DPP9-kd cells compared to the control parental HeLa cells (*Figure 5I and J*), suggesting a functional cross-talk between DPP9 and Cbl.

We further tested the outcomes of Syk destabilization by DPP9 in B cells. Western blot analysis, activity assays and indirect immunofluorescence verify the expression of DPP9 in these cells (*Figure 6—figure supplement 1*). Next, PLA analysis shows that the binding of FLNA to Syk and DPP9 is conserved in DG-75 cells (*Figure 6A and B*). Importantly, clear PLA signals corresponding to an interaction between DPP9 and Syk were also observed in these cells (*Figure 6C and D* and *Figure 6—figure supplement 2*). Of note, DPP8, which shares 60% homology to DPP9, did not appear to interact with Syk, although it is expressed in these cells. Similarly, no interaction above background level was observed between Syk and the membrane protease DPPIV, which shares 26% homology with DPP9 (*Figure 6C and D*).

Next, CHX chase experiments were performed to analyse the half-life of endogenous Syk in stimulated (BCR-engaged) and unstimulated DG-75 cells. BCR signalling was elicited by treating cells with anti-BCR F(ab)$_2$ when adding the CHX. DPP9 and Tubulin remained unaltered over the respective chase period and were assessed as a loading control (*Figure 6E*). Whereas Syk was stable in unstimulated B cells over a time period of 6 hr, upon stimulation of the BCR, Syk was degraded with a half-life of 2 hr (+ stimulation, *Figure 6E*). Syk half-life in stimulated DG-75 cells is shorter than the previously published half-life of Syk in stimulated natural killer cells where it is estimated to be 8 hr (*Paolini et al., 2001*). As expected, a clear stabilization of Syk was observed in stimulated DG-75 cells which were treated with the proteasome inhibitor MG132 (*Figure 6F and G*). Notably, consistent with our findings showing stabilization of Syk in DPP9-kd HeLa cells, treatment of stimulated DG-75 cells with the DPP8/9 inhibitor 1G244 resulted in a clear stabilization of Syk (*Figure 6F and G*).

Next, we applied specific antibodies recognizing Syk phosphorylated on Y323, which was previously shown to be important for binding to Cbl (*Yankee et al., 1999*; *Lupher et al., 1998*). CHX chase experiments on BCR stimulated cells show clear signals corresponding to the phosphorylation

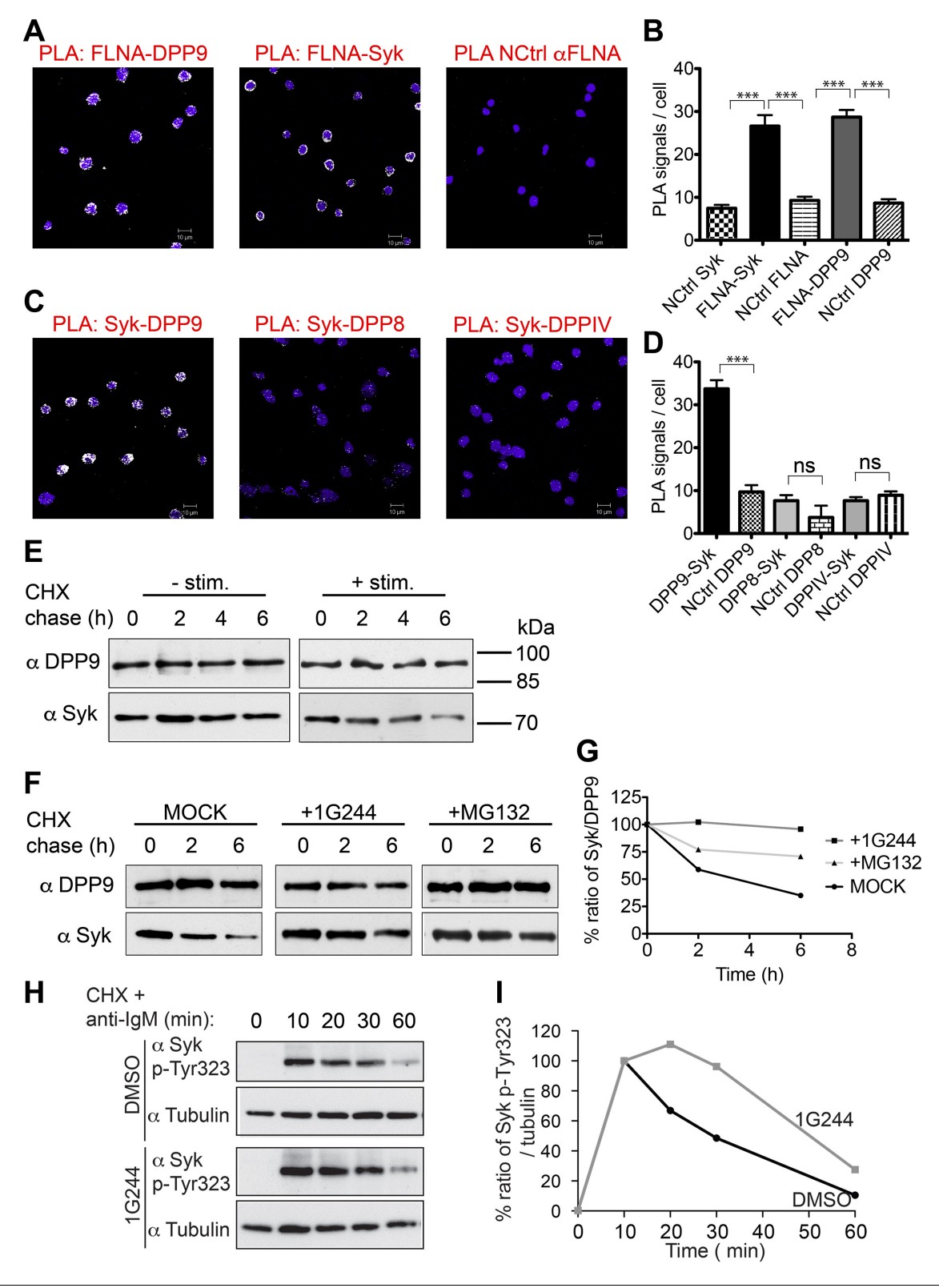

**Figure 6.** DPP9 regulates the stability of Syk in human Burkitt's lymphoma B cells. (**A** and **B**) PLA showing that the interactions of FLNA with Syk and DPP9 are conserved in human DG-75 B cells. Each PLA interaction is shown here as a white dot, nuclei were visualized by using HOECHST. Control reactions (NCtrl) were performed with only one primary antibody (αSyk, αFLNA or αDPP9). Shown are representative images and quantifications of at least three independent PLA experiments. The number of PLA signals per cell were quantified in a blinded manner using the Duolink ImageTool
*Figure 6 continued on next page*

*Figure 6 continued*

software (SIGMA). Signals of more than 80 cells were quantified for each condition respectively. Data are represented as mean ± SEM. Statistical analysis was carried out by an unpaired two-tailed t test (***p<0.0001). (C and D) PLA in DG-75 cells showing that Syk interacts specifically with DPP9 but not with its homologs DPP8 and DPPIV. Control reactions (NCtrl) cells were treated with one primary antibody only: αDPP9, αDPP8 or αDPPIV. Shown are quantifications of the PLA DPP9-Syk, DPP8-Syk and DPPIV-Syk in DG-75 cells from three independent experiments. Data are represented as mean ± SEM. The number of PLA signals per cell were quantified in a blinded manner using the Duolink ImageTool software (SIGMA). Signals of more than 100 cells were quantified for each condition respectively. Statistical analysis was carried out by an unpaired two-tailed t test (***p<0.0001; n.s = not significant). (E) CHX chase experiment showing reduced stability of endogenous Syk upon stimulation of the BCR. Human DG-75 cells were stimulated with 12 µg/ml F(ab')$_2$ fragment goat-anti-human IgG+IgM (+ stim), or left untreated (- stim), and simultaneously subjected to CHX chase. DPP9 was analysed as a loading control. Shown is one representative result of at least three independent experiments. (F) CHX chase experiments showing that the stability of endogenous Syk in stimulated DG-75 cells, is determined by the proteasome and by DPP9. DG-75 cells were treated either with the DPP8/9 inhibitor 1G244 (10 µM), with the proteasome inhibitor MG132 (100 µM) or with DMSO for control (MOCK). Cell lysates were analysed for protein levels of Syk and of DPP9 for loading control by Western blotting. Shown is one representative result of at least three independent experiments. (G) Quantification of the Western blot results shown in (F). The ratio of Syk/DPP9 at time 0 hr was normalized to 100%. For signal quantification GelQuant.NET software provided by biochemlabsolutions.com was used. (H) CHX chase experiment assaying the stability of endogenous phosphorylated Syk (p-Y323) in stimulated DG-75 cells upon treatment with the DPP8/9 inhibitor 1G244 (10 µM) or with DMSO for control (MOCK). Tubulin was assayed as loading control. Shown is one representative result of at least three independent experiments. (I) Quantification of the Western blot results shown in (H). The ratio of Syk p-Y323/tubulin at time 10 min was normalized to 100%. For signal quantification GelQuant.NET software provided by biochemlabsolutions.com was used.

The following figure supplements are available for figure 6:

**Figure supplement 1.** DPP9 expression, activity and interaction with Syk in the human DG-75 cells.

**Figure supplement 2.** Controls for the DPP9-Syk PLA in DG-75 cells.

of Syk on Y323 upon stimulation of DG-75 cells (*Figure 6H and I*), which clearly dropped in intensity over time. Notably, the intensities of signals corresponding to Syk pY323 did not drop as rapidly in 1G244 treated cells. Instead, we observed an increase in signal intensity up to 20 min post BCR stimulation, suggesting that during this time more Syk is phosphorylated on Y323, but degradation is prevented due to inhibition of DPP9. Taken together, these results show a central role for DPP9 in determining the stability of Syk after BCR-engagement, and suggest that DPP9 functions upstream of Cbl.

## DPP9 is a negative regulator of Syk signalling in B cells

In addition to Y323, engagement of the BCR initiates the phosphorylation of Syk on multiple sites within a short time (*Furlong et al., 1997*; *Bohnenberger et al., 2011*; *Satpathy et al., 2015*). Phosphorylation serves both to modulate Syk catalytic activity and to modify its interactions with other proteins (*Mócsai et al., 2010*; *Geahlen, 2009*; *Krisenko and Geahlen, 2015*). Remarkably, inhibition of DPP9 in stimulated B-cells resulted also in a prolonged phosphorylation of Syk on Y352 (*Figure 7A and B*), which is important both for the catalytic activity of Syk and for the activation of the proximal target Phospholipase C-γ2 (PLCγ2) (*Tsang et al., 2008*; *Law et al., 1996*). Similar to Syk pY323 (*Figure 6H and I*), signals for Syk-pY352 continued to increase up to 20 min in 1G244 treated cells, but not in the control cells (*Figure 7A and B*). In total, the reduction in the signal intensities for Syk p-Y352 was slower in DG-75 cells treated with 1G244 compared to the mock treated control cells (*Figure 7A and B*), suggesting that DPP9 cleaves active phosphorylated Syk.

Next, we analysed the activity of PLCγ2, which is phosphorylated by Syk upon BCR stimulation resulting in Ca$^{2+}$ release from the endoplasmic reticulum (*Engelke et al., 2007*). Remarkably, application of 1G244 to DG-75 cells triggered a rapid increase in Ca$^{2+}$ fluxes, which occurred before the stimulation of the BCR (*Figure 7C*). The levels of Ca$^{2+}$ reached a plateau within less than 3 min after application of 1G244, in DG-75 cells, and were less elevated upon BCR engagement compared to control cells. Similar results were also observed in Ramos cells, a second Burkitt's lymphoma cell line (*Figure 7D*).

The rapid increase in calcium fluxes in response to DPP9 inhibition independently of BCR stimulation was surprising, since Syk appears to be stable in unstimulated B cells (*Figure 6E*). Furthermore, in the absence of BCR stimulation, the steady state levels of Syk were comparable in cells treated

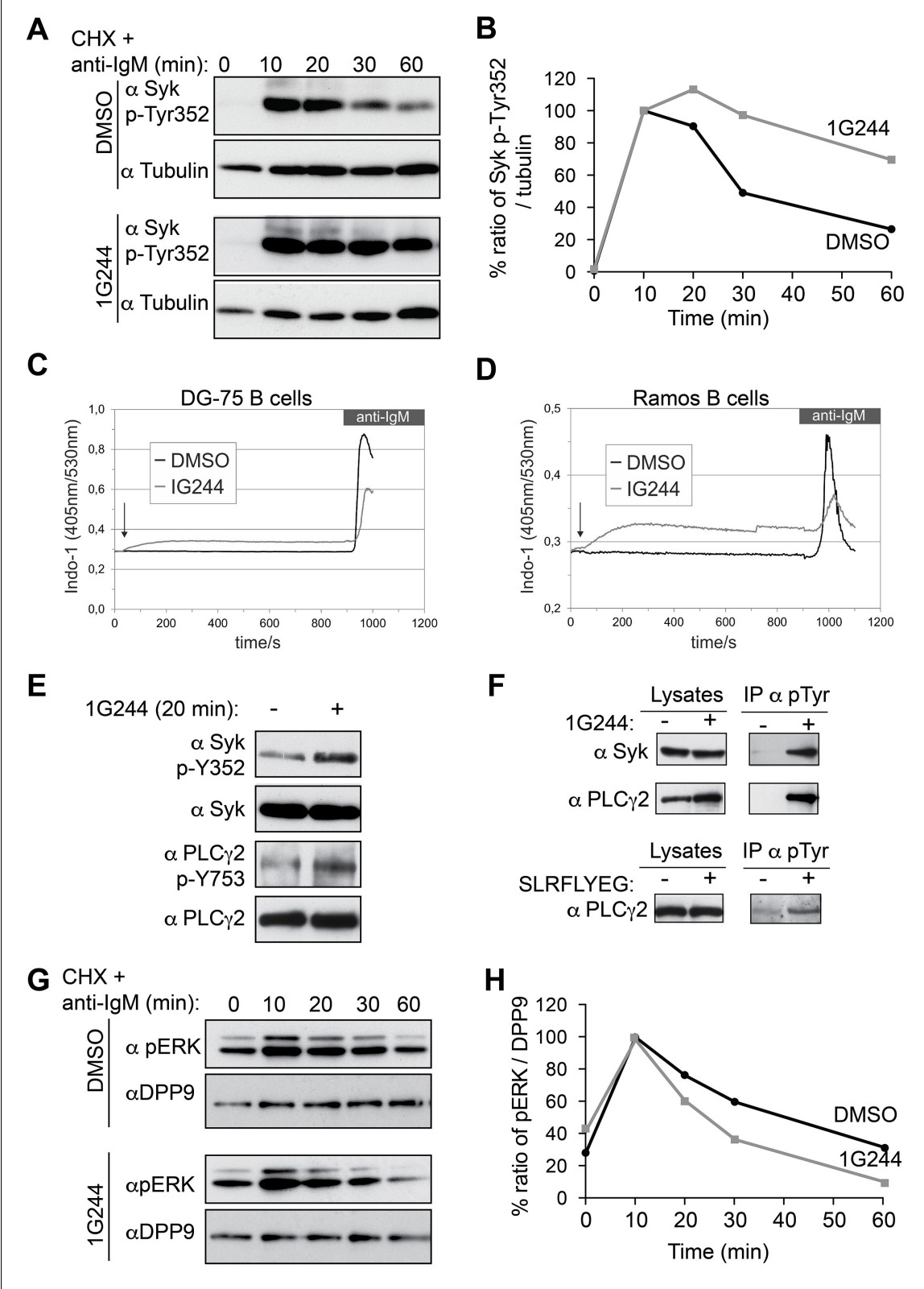

**Figure 7.** DPP9 targets phosphorylated Syk for degradation thus influencing Syk signalling in B cells, (**A**) Higher levels of endogenous active Syk (phosphorylated on Y352) in stimulated DG-75 cells treated with the DPP8/9 inhibitor 1G244 compared to the mock (DMSO) treated cells. 1G244 (10 µM) was added at the same time of BCR stimulation (time 0). Tubulin was assayed for loading control. Shown is a representative result of at least three independent pulse chase experiments. (**B**) Quantification of the Western blot results shown in (**A**). The ratio of Syk p-Y352/tubulin at time 10 min was

*Figure 7 continued on next page*

*Figure 7 continued*

normalized to 100%. For signal quantification GelQuant.NET software provided by biochemlabsolutions.com was used. (C) Inhibition of DPP9 in DG-75 cells leads to increased $Ca^{2+}$ mobilization, which is not dependant on BCR stimulation. Shown are flow cytometric $Ca^{2+}$ profiles after the addition of 10 μM DPP8/9 inhibitor 1G244 or DMSO for control (marked by an arrow). To monitor $Ca^{2+}$ mobilization upon BCR stimulation in either 1G244-treated or control cells, cells were treated with 10 μg/ml $F(ab)_2$ goat-anti-human IgM. (D) Same as in (C) using Ramos cells as a second B cell line. (E–F) In the absence of BCR stimulation, DPP9 inhibition leads to higher basal levels of phosphorylated Syk and its down stream effector protein PLCγ2. (E) Western blotting analysis of DG-75 cells treated with 1G244 in the absence of BCR stimulation. Lysates were analysed with antibodies specific against phosphorylated Syk (p-Y352) or phosphorylated PLCγ2. For loading control lysates were analysed with antibodies recognizing unmodified Syk and PLCγ2. (F) Cells were treated for 20 min with 1G244 (10 μM) or DMSO for control. Alternatively, cells were treated for 30 min with the allosteric DPP9 inhibitor SLRFLYEG peptide complexed with the carrier peptide (pep-1). Control cells were treated with the carrier peptide only. Following inhibitor treatment, cells were lysed and subjected to immunoprecipitation assays against Phospho-Y. Eluted proteins were analysed for Syk and PLCγ2 levels by Western blotting. Total protein levels in cell lysates were monitored for control. (G) Lower levels of phosphorylated ERK1/2 (both bands) are detected in the 1G244 treated DG-75 cells compared to mock (DMSO) treated cells. 1G244 (10 μM) was added prior to BCR stimulation (time 0). DPP9 was assayed for loading control. Shown is a representative result of at least three independent pulse chase experiments. (H) Quantification of the Western blot results shown in (G) as described in (B).

with 1G244 and in mock treated cells (no 1G244) (*Figure 7E*). Similar results were observed for PLCγ2. However, as we tested for the levels of active syk and PLCγ2, we found that the phosphorylation levels of Syk (Syk-pY352) and PLCγ2 (PLCγ2-pY753) were higher in the 1G244 treated cells (*Figure 7E*). These observations are in line with the increased $Ca^{2+}$ fluxes under these conditions (*Figure 7C and D*). Immunoprecipitation assays of DG-75 cell lysates with antibodies against phospho-tyrosine (p-Y) (*Figure 7F*) also revealed higher levels of phosphorylated Syk and PLCγ2 upon DPP9 inhibition with 1G244 (*Figure 7F*), independently of BCR stimulation. Consistently, an increase in the steady state phosphorylation levels of PLCγ2 on pY753 was also observed in DG75 cells treated with the allosteric inhibitor of DPP9 SLRFLYEG in the absence of BCR engagement (*Figure 7F*).

To further test the biological consequences of DPP9 activity, we analysed the effect of DPP9 inhibition on the extracellular signal-related kinase (ERK) 1/2 which is a distal downstream target of Syk (*Slack et al, 2007*). We therefore incubated DG-75 cells with 1G244 prior to BCR stimulation and subsequently analysed ERK1/2 phosphorylation. While BCR engagement under DPP9 inhibition revealed a more transient ERK1/2 phosphorylation profile compared to control cells, it is important to note that the levels of ERK1/2 phosphorylation in resting cells were slightly augmented in cells that were treated with 1G244 (*Figure 7 G and H*). These results are in line with the finding that DPP9 inhibition in unstimulated cells stabilizes an active phosphorylated form of Syk, resulting in 'signal leakage' as measured by higher levels of phosphorylated PLCγ2 forms and increased $Ca^{2+}$ fluxes leading to reduced sensitivity to BCR engagement. Taken together, these results suggest that DPP9 is an integral negative regulator of Syk signalling by terminating Syk activity through processing the amino terminus of the kinase.

## Discussion

### DPP9 – a negative regulator of Syk signalling

Here, we report the identification of Syk as a novel endogenous substrate of DPP9. Being a central component of the B cell signalling, Syk initiates the phosphorylation of a cascade of downstream components, ultimately modulating cellular metabolism, cell migration, cell proliferation, apoptosis and gene transcription events (*Mócsai et al., 2010*; *Lowell, 2011*). Syk is activated following BCR engagement, and is negatively regulated by phosphatases such as Shp-1 and PTPROt (*Dustin et al., 1999*; *Alsadeq et al., 2014*; *Chen et al., 2006*), and by the E3 ubiquitin ligase Cbl (*Paolini et al., 2001*; *Joazeiro et al., 1999*; *Rao et al., 2001a*; *Sohn et al., 2003*; *Lupher et al., 1998*; *Ota and Samelson, 1997*). Here, we present evidence for the role of DPP9 as a novel negative regulator of Syk that determines Syk stability. DPP9 inhibition results in increased half-life of Syk and higher levels of active phosphorylated Syk pY352, suggesting that DPP9 targets the active form of Syk for degradation.

Furthermore DPP9 inhibition results in a prolonged accumulation of Syk phosphorylated on Y323, which is a critical residue for interaction with Cbl (*Yankee et al., 1999*; *Lupher et al., 1998*). Moreover, the interaction between Cbl and Syk is reduced to background level upon DPP9 silencing. Taken together, these results strongly suggest a cross talk between DPP9 and Cbl, in which cleavage of Syk by DPP9 is an up stream event to Syk ubiquitination by Cbl.

Surprisingly, also in the absence of BCR-engagement, DPP9 inhibition leads to higher steady state levels of phosphorylated Syk (pY352) and PLCγ2 (pY753) as well as to a rapid mobilization of $Ca^{2+}$ from the ER. These results show that inhibition of DPP9 prior to BCR engagement leads to a 'leakage' in Syk signalling, and consequently reduces the response of these cells to BCR-stimulation as measured by lower levels of $Ca^{2+}$ release upon BCR-engagement probably due to the depletion of $Ca^{2+}$ in the resting stage. Consistently, we observed reduced BCR mediated phosphorylation of ERK1/2 upon DPP9 inhibition in these cells.

In addition to activated BCR signalling, which relays on the binding of antigens to the BCR, B cells also possess the so-called 'tonic' signalling, which maintains a constitutive baseline signal in B cells. Tonic signalling occurs in the absence of antigen binding, but nonetheless is dependent on the presence and activity of the BCR and some BCR components such as Syk, PLCγ2 and ERK1/2 (*Monroe, 2006*; *Wienands et al., 1996*; *Shaffer and Schlissel, 1997*). The molecular details regarding this pathway are only emerging but it is clear that tonic signalling is essential for survival of B cells, including Burkitt lymphomas (*Corso et al., 2016*). We suggest that by targeting Syk pY352 for degradation, DPP9 maintains low levels of activated Syk sufficient for tonic signalling, and thus can influence the strength of the BCR signalling in response to antigen engagement. By targeting Syk for degradation after BCR engagement, DPP9 can also influence the duration of the response to antigen BCR binding (*Figure 8A and B*).

Several reports link DPP9 to various cellular processes, including cell migration, metabolism, cell proliferation and apoptosis (*Zhang et al., 2015*; *Yao et al., 2011*; *Matheeussen et al., 2013*; *Spagnuolo et al., 2013*; *Chen et al., 2016*; *Han et al., 2015*). It remains to be shown whether some of the effects caused by DPP9 silencing are due to stabilization of Syk in these cells. Since Syk appears to act as a negative regulator of specific tumours (*Coopman and Mueller, 2006*), inhibition of DPP9 to stabilize Syk in these cells may reflect a future approach for tumour therapy.

## DPP9 - a novel upstream regulator of the N-end rule pathway

Our data show that processing of Syk by DPP9 exposes a neo Syk N-terminus with a serine residue in position 1 (*Figure 8C*). Silencing (DPP9-kd) or inhibition of DPP9, or mutation of the DPP9 cleavage site in Syk (SykD2A) result in an increased stability of Syk. The significance of the exposed Ser for Syk stability was demonstrated by mutagenesis of this residue to a Gly or Val, resulting in a longer half-life of these variants. Therefore, we conclude that Syk is a novel substrate of the N-end rule pathway and DPP9 a novel peptidase in this pathway (*Figure 8C*, and see more below). Of note classical primary N-degron are positively charged or bulky hydrophobic amino acids. According to this definition, Ser, Gly and Val are not destabilizing amino acids per se (*Bachmair and Varshavsky, 1989*), however they can form an N-degron upon N-terminal acetylation (Ac/N-degrons) (*Hwang et al., 2010b*; *Shemorry et al., 2013*; *Park et al., 2015*). Interestingly, the difference in the stability of Syk wild-type, SykS3V and SykS3G mirrors the N-acetylation efficiency of these residues (*Arnesen et al., 2009*).

DPP9 inhibition results in a prolonged accumulation of Syk phosphorylated on Y323, which is a critical residue for interaction with the E3 ligase Cbl (*Yankee et al., 1999*; *Lupher et al., 1998*). Moreover, the interaction between Cbl and Syk is reduced to background level upon DPP9 silencing. Taken together, these results strongly suggest a cross talk between DPP9 and Cbl, in which cleavage of Syk by DPP9 is an up stream event to Syk ubiquitination by Cbl. The functional link between Cbl and DPP9 suggests a novel role for Cbl in the N-end rule pathway, further expanding the growing list of ubiquitin E3 ligases which includes Doa10/Teb4 and Not4 that recognize non-primary N-degrons (*Hwang et al., 2010b*; *Shemorry et al., 2013*; *Park et al., 2015*).

Proteases such as calpains, separins, metalloproteases and caspases are linked to the N-end rule pathway, since they produce protein fragments that are degraded by this pathway (*Rao et al., 2001b*; *Brower et al., 2013*; *Piatkov et al., 2014*; *Liu et al., 2016*). Knowledge regarding the participation of aminopeptidases in the N-end rule pathway is limited to methionine aminopeptidases (MetAPs) (*Varshavsky, 2011*). These remove a single methionine from the amino terminus of a

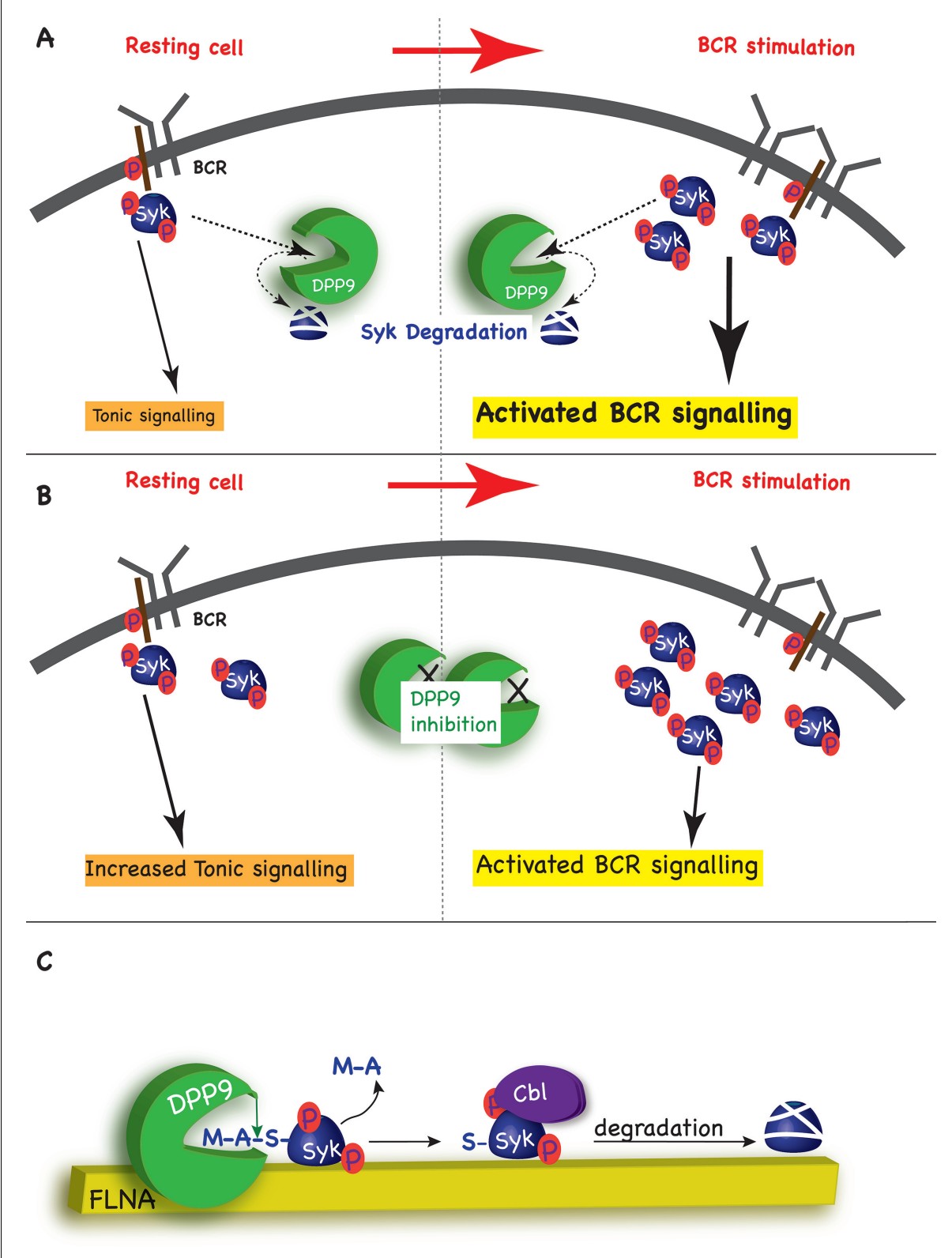

**Figure 8.** Model - DPP9 is a negative regulator for Syk signalling, targeting active Syk for degradation by the N-end rule pathway. (**A**) Model: DPP9 regulates Syk signalling. By maintaining low levels of activated Syk in resting B cells DPP9 controls tonic signalling and preserves signalling capacity for the processes induced by BCR engagement. By targeting Syk for degradation after BCR engagement, DPP9 can also influence the duration of the response to antigen BCR binding. (**B**) Reduced DPP9 activity e.g by inhibition results in higher levels of active Syk in non-stimulated cells. This results in

*Figure 8 continued on next page*

Figure 8 continued

elevated Tonic signalling ('signal leakage'), consequently leading to a lower response upon BCR engagement. (C) Model: DPP9 targets Syk for degradation by the N-end rule pathway. In this model, FLNA acts as a recruitment platform for binding of Syk and DPP9. FLNA then supports the cleavage of Syk by DPP9, by increasing the local concentration of Syk and DPP9, stabilizing the interaction between Syk and DPP9, or by optimizing the orientation of DPP9 and Syk. The interaction between DPP and Syk leads to the processing of Syk N-terminus which removes the dipeptide MA, and exposes a neo Syk N-terminus with serine in position 1 is exposed. Subsequently the E3 ligase Cbl binds to Syk p-Y323 initiating its ubiquitintaion and degradation by the proteasome.

protein substrate leaving the protein otherwise intact. Of note, cleavage by MetAPs does not directly form an N-degron, since MetAPs remove the initiator methionine from substrates where the adjacent residue (P1') is not destabilizing: Ser, Pro, Gly, Thr, Ala, Val or Cys (*Bachmair and Varshavsky, 1989*; *Xiao et al., 2010*; *Addlagatta et al., 2005*; *Frottin et al., 2006*). Nonetheless, MetAPs are considered an integral part of the N-end rule pathway, because these residues may be converted to Ac/N-degrons (*Hwang et al., 2010b*), albeit to different efficiencies. Our data place DPP9 as a second integral aminopeptidase of the N-end rule pathway. In contrast to MetAPs, DPP9 cleavage is permissive towards more amino acids in the P1' position (Xaa-Ala/Pro-Zaa), and is only inhibited if the residue after the cleavage site (Zaa) is Pro (*Geiss-Friedlander et al., 2009*). Therefore DPP9 may directly produce an N-degron not requiring further modification (*Figure 8C*). Additionally, DPP9 can cleave substrates with any amino acid in the P2 position (Xaa) apart from Asp or Glu, greatly increasing the number of potential substrates, including proteins undergoing progressive processing (*Geiss-Friedlander et al., 2009*; *Connolly et al., 2008*).

Whereas MetAPs function largely in a co-translational manner, our data strongly suggest that DPP9 acts in a post-translational manner. Consequently, processing by DPP9 and MetAPs and funnelling of their products to the N-end rule pathway may be regulated by different signals. MetAP substrates, which were processed and acetylated at the N-terminus co-translationally, can be protected from degradation by masking of the N-degron, e.g. via protein interactions, and exposed upon changes in protein stoichiometry (*Hwang et al., 2010b*; *Shemorry et al., 2013*). On the other hand, DPP9 substrates may be processed only upon a specific post-translational cue. For Syk, our data suggest increased cleavage by DPP9 upon BCR stimulation indicated by a shorter half-life of Syk in stimulated cells, and clear stabilization by 1G244 treatment comparable to that observed with the proteasome inhibitor MG132. This increased processing of Syk upon BCR stimulation may depend for example on Syk phosphorylation.

Additionally, we found that although DPP9 interacts directly with Syk, in cells this interaction requires FLNA, suggesting that FLNA may serve as an interaction platform linking DPP9 and Syk (*Figure 8C*). Consequently, FLNA may support Syk processing e.g. by increasing the local concentration of Syk and DPP9, stabilizing the complex formation or by optimizing the orientation of DPP9 and Syk for efficient cleavage. In this sense, the function of FLNA may parallel that of F-box proteins in the ubiquitin ligase SCF complexes, which bind to substrates and thus support their ubiquitination by ubiquitin conjugating enzymes (*Edward and Kipreos, 2000*). Since FLNA interacts with multiple proteins, it is tempting to speculate that it acts as a general recruitment factor targeting DPP9 to additional potential substrates for cleavage.

In conclusion, here we identify DPP9 as a novel upstream component of the N-end rule pathway. We suggest that depending on the substrate and the exposed residue in the neo N-terminus, DPP9 may cooperate with Cbl or other E3 ligases that then ubiquitinate the substrate resulting in proteasomal degradation.

## Materials and methods

### Cell culture, transfection and silencing

HeLa DPP9 silenced cells (DPP9-kd) were custom made for us by GenScript. Culturing of HeLa DPP9 silenced cells (DPP9-kd) and the corresponding HeLa parental wt cells were cultured at 37°C and 5% $CO_2$ in Dulbecco's modified Eagle's medium supplemented with 10% fetal bovine serum, 2 mM L-glutamine, 100 U/ml penicillin and 100 μg/ml streptomycin. To maintain the selection pressure on

the DPP9-kd cells, 1.5 µg/ml puromycin (Sigma-Aldrich, Germany) was added to the growth medium. Human DG-75 and Ramos B cells (purchased ATCC, Germany) were cultured in VLE RPMI medium (Biochrom, Germany) supplemented with 3 mM glutamine, 10% heat-inactivated FCS, 100 mM sodium pyruvate and 100 U/ml penicillin and 100 µg/ml streptomycin. All cells are routinely tested for mycoplasma (by GATC Biotech, Germany).

## Plasmids

C-terminally tagged full-length Syk was generated via PCR using a Syk-containing plasmid and was subsequently cloned into pcDNA3.1+ (using the BamHI and NotI restriction sites). Single amino acid exchanges at the N-terminus were introduced using primers for site-directed mutagenesis. All plasmids were sequenced before usage.

## Transfections and silencing

For overexpression experiments, HeLa (P4) cells were transfected using the calcium-phosphate method at a cell confluency of 50–60% and analysed 48 hr after transfection. Control cells were transfected with a GFP plasmid (MOCK). For silencing of DPP9 HeLa cells were transfected essentially as previously described in (Geiss-Friedlander et al., 2009). SiRNA oligonucleotides for Syk were purchased from Santa Cruz Biotechnology Inc. Germany, siRNA oligonucleotides targeting FLNA were purchased from Invitrogen. For control, cells were transfected with a non-targeting siRNA oligonucleotide (Invitrogen). Cells were analysed 72 hr after transfection.

## Antibodies

For Western blotting rabbit anti-DPP9 (#ab42080, 1:1000) was purchased from Abcam, England. We produced the goat anti-DPP9 antibodies (1:1000) by injecting recombinant DPP9-short protein to a goat. Antibody specificity is shown in *Figure 1—figure supplement 1A*. Rabbit anti-Syk (#sc-1077, 1:1000) and the corresponding blocking peptide rabbit anti-Syk (#sc-929), mouse anti-Tubulin (#sc-32293, 1:1000), rabbit anti-GFP (#sc-8334, 1:1000), and mouse anti-GST (#sc138, 1:1000) antibodies were obtained from Santa Cruz Biotechnology Inc. Mouse anti-FLAG (#f1804, 1:1000) was purchased from SIGMA and mouse anti-FLNA (#MAB1680, 1:1000) from Millipore, Germany. Mouse anti-pTyr 100 (#9411), rabbit anti-phospho-ERK1/2 (#4370 1:2000), rabbit anti PLCγ2 (1:2000, Santa Cruz Biotechnology, #sc-9015), rabbit anti p-PLCγ2 (Y753) (1:500, Santa Cruz Biotechnology, #sc-101785), rabbit anti Syk (#2712, 1:1000) and rabbit anti-phospho-Syk (Y323, #2715, 1:1000; Y352, #2701, 1:1000) were obtained from Cell Signalling Technology, The Netherlands. The enhanced chemiluminescence system (Millipore) was used for visualization of proteins on the membranes. For signal quantification GelQuant.NET software provided by biochemlabsolutions.com was used.

In immunofluorescence studies and in situ Proximity ligation assays (PLAs) the following antibodies were used: self-generated goat anti-DPP9 (1:10–20), mouse anti DPPIV (1:50–100; Santa Cruz Biotechnology, #sc-19607), mouse anti DPP8 (1:50–100, Santa Cruz Biotechnology, #sc-37699), rabbit anti-Syk (1:100–1:200), goat anti-Cbl (C-15) (1:60; Santa Cruz Biotechnology, #sc-170-G), mouse anti-FLNA (1:100), rabbit anti-FLNA (1:100, NB100-58812; Novus Biologicals, Germany) and mouse anti-HA (1:400, , MMS-101P; Covance Germany). Secondary antibodies for immunofluorescence: donkey anti-mouse Alexa-Fluor-488, donkey anti-goat Alexa-Fluor-594 and donkey αnti-rabbit Alexa-Fluor-594, all at a dilution of 1:500; purchased from Molecular Probes. For PLAs the Duolink In situ PLA kit was purchased from Sigma-Aldrich. The PLA Probes (oligonucleotide-conjugated secondary antibodies) were used in the combinations: anti-Rabbit PLUS (#DUO92002) with either a Goat MINUS (#DUO92006) or a Mouse MINUS (#DUO92004), and a Goat MINUS (#DUO92006) with a Mouse PLUS (#DUO92001); the probes were diluted 1:5. For staining of actin filaments in fixed cells CytoPainter Phalloidin-iFluor 488 (1:650, #ab176753, Abcam) was used. For staining of actin filaments in fixed cells CytoPainter Phalloidin-iFluor 488 (1:650, #ab176753, Abcam) was used.

## Inhibitors and peptides

The DPP8/9-specific inhibitor 1G244 was purchased from AK Scientific, Inc. (Union City, CA), the proteasome inhibitor MG132 was purchased from Enzo Life Sciences. All peptides including the SLRFLYEG peptide were custom-made by GenScript (Hong Kong), to > 80% purity.

## Yeast two-hybrid screen

A human placenta library was screened using full-length DPP9-short as bait. The assay was performed by Hybrigenics (Paris, France).

## Indirect immunofluorescence

For immunofluorescence, HeLa cells grown on coverslips were fixed with 4% formaldehyde in PBS containing 10 µg/ml Hoechst 33,258 (Molecular Probes) for staining of the nuclei. Subsequently, cells were permeabilized with 0.2% Triton-X-100 in PBS, washed and blocked for 10 min in blocking buffer (2% BSA in PBS). For IF of DG-75 cells, $3*10^5$ cells per slide were resuspended in PBS, added to the cover slip and incubated for 30 min at RT to allow attachment of the cells to the surface. Cells were fixed with 1% formaldehyde in PBS containing 10 µg/ml Hoechst 33,258 for staining of the nuclei. Subsequently, cells were permeabilized with 0.02% Triton-X-100 in PBS, washed and blocked for 30 min in blocking buffer. Both cell lines were incubated with primary antibodies for 90 min at 37°C. Following a PBS wash, cells were then incubated for 45 min at room temperature with the respective secondary antibodies. Cells were washed with PBS and water, and mounted in fluorescent mounting medium (DAKO). Control samples were treated with secondary antibody only, to estimate background staining. Cells were analysed and images were taken using a LSM 510-Meta confocal microscope, oil immersion objective 63x/1.3 (Carl Zeiss MicroImaging, Inc). Images were processed using the LSM image Browser (Carl Zeiss MicroImaging, Inc) and Adobe Photoshop.

## Proximity ligation assay (PLA)

PLA was performed using the DUOLINK In Situ PLA Kit from Sigma Aldrich, according to the manufacturer's protocol. In short, HeLa cells and DG-75 cells were grown on coverslips and fixed as described above. To investigate consequences of DPP9 inhibition prior to fixation cells were treated either with 10 µM 1G244 or DMSO as MOCK for 5 or 30 min or alternatively with the allosteric peptide inhibitor SLRFLYEG. For entry into the cells the SLRFLYEG peptide was incubated with the carrier peptide (pep1) to form the SLRFLYEG-pep1 complexes for 30 min or 1 hr at 37°C (as described in *Pilla et al. (2013)*). Control assays contained pep1 alone. Both cell lines were then incubated with primary antibodies, 90 min at 37°C and actin filaments were simultaneously counterstained with CytoPainter Phalloidin-iFluor 488 Reagent (Abcam - #ab176753, 1:650 dilution). In case of a control for antibody specificity, the antibody was pre-incubated with the corresponding blocking peptide (1 µg/ml) for 30 min at RT before addition to the cover slips. To estimate background staining in each experiment control slides were treated with one primary antibody only. After careful wash with PBS, cells were treated with the PLA probes according to the manufacturer's protocol. Cells were mounted in DAKO fluorescent mounting medium and analysed using a LSM 510-Meta confocal microscope, oil immersion objective 63x/1.3 (Carl Zeiss MicroImaging, Inc). Taken images were processed using the LSM image Browser (Carl Zeiss MicroImaging, Inc) and subsequently analysed using the Duolink ImageTool (SIGMA).

## Surface plasmon resonance (SPR)

Binding experiments were performed on a Reichert SPR Biosensor (SR 7500 C, Reichert Instruments), using $Ni^{2+}$ chelator chips (NiHC 1000 m, Xantec Bioanalytics, Duesseldorf). The surface was initially purged of contaminants injecting 0.5 M EDTA, pH 8.5, followed by equilibration steps washing the surface with immobilization buffer (20 mM Hepes, 150 mM NaCl, 0.005% Tween 20, pH 7.4) at a flow rate of 30 µl/min. The left channel of the chip surface (ligand channel) was charged with $Ni^{2+}$ using 0.3 M $NiSO_4$. A 200 nM solution of His-tagged, affinity purified DPP9 was subsequently injected at a flow rate of 30 µl/min. The ligand (DPP9) was immobilized to a response of 2500 to 3500 µRIU. The right channel was kept empty and used as the first reference. Synthesized peptides correlating the first 31 amino acids of the Syk N-terminus were used as the analyte. A serial dilution of the peptide was injected over the chip surface at concentrations of: 16 µM, 8 µM, 4 µM, 2 µM, 1 µM, 0.5 µM, 0.25 µM and 0.125 µM. For each analyte sample twice as many buffer injections were performed, which were later on used as a second reference. After each binding experiment, containing two buffer injections and one analyte injection, the chip surface was washed with EDTA (0.5 M pH 8.5) and new ligand (DPP9) was immobilized as described above. All binding experiments were carried out at a flow rate of 40 µl/min in SPR running buffer (20 mM Hepes, 150 mM NaCl, 0.005%

Tween 20, 50 µM EDTA, pH 7.4). The response of each analyte sample was doubly referenced with the response obtained from the reference channel (right channel) and the response obtained by injecting buffer using Scrubber version 2.0c. Equilibrium binding analysis was performed using Graph Pad Prism 6.0.

## Recombinant protein expression and purification

Recombinant DPP9-short was expressed in SF9 insect cells and purified essentially as described in (*Nakamura et al., 2007*). Full-length FLNA wild-type and variants were purified as described in (*Yue et al., 2013*). For expression of GST-FLNA fragments 5–7: FLNA repeats 5–7 or 6–7 in pGEX4T1 were transformed into *Escherichia coli* BL21 (Stratagene). Cells were grown to A600 0.6 and induced with 0.1 mM isopropyl 1-thio-β-D-galactopyranoside for 3 hr at 30°C. All following buffers were supplemented with protease inhibitors (1 µg/ml each of leupeptin, pepstatin, and aprotinin), and 1 mM dithiothreitol (DTT). Cells were harvested and resuspended in lysis buffer (50 mM Tris-HCl, pH 8.0, 100 mM NaCl, 1 mM EDTA, 1 mM EGTA). Cells were disrupted using an EmulsiFlex (Avestin) and centrifuged for 1 hr at 100,000 ×g. The supernatant was incubated with 1 ml Gluthion-Sepharose (Macherey-Nagel) for 1 hr at 4°C. Beads were washed at 4°C with binding buffer (50 mM Tris-HCl, pH 8.0, 300 mM NaCl, 1 mM EDTA, 1 mM EGTA), supplemented with protease inhibitors and 1 mM DTT. Proteins were eluted with elution buffer (20 mM glutathione in 50 mM Tris-HCl, pH 8.0, 300 mM NaCl, 1 mM EDTA, 1 mM EGTA) supplemented with protease inhibitors and 1 mM DTT and further purified using an Äkta Purifier (GE Healthcare) equipped with a Superdex 75 size exclusion column (GE Healthcare) in Transport buffer (20 mM Hepes, pH 7.3, 110 mM potassium acetate, 2 mM Mg acetate, 1 mM EGTA) supplemented with protease inhibitors and 1 mM DTT.

## Kinetic assays

To measure DPP activity in DG-75 cells, $2*10^7$ cells were resuspended in 2 ml of RPMI complete medium containing either 10 µM 1G244 or DMSO (MOCK) and incubated for the corresponding times (5 min, 30 min) at 37°C. The reaction was stopped with 20 ml ice-cold PBS and cells were pelleted for 5 min at 500 g. Subsequently, cells were washed with 10 ml ice-cold PBS and were shock-frozen in liquid $N_2$. For activity measurements, cell pellets of the respective cell line were lysed in TB buffer (20 mM HEPES/KOH, pH 7.3, 110 mM potassium acetate, 2 mM magnesium acetate, 0.5 mM EGTA) supplemented with 0.02% Tween 20 and 1 mM DTT, centrifuged for 20 min at 55,000 rpm, 4°C. Next, 5 µg cell lysate was incubated with either 250 µM Gly-Pro-AMC (GP-AMC) or 50 µM Arg-AMC (R-AMC), fluorescence release was measured using the Appliskan microplate fluorimeter (Thermo Scientific) with 380 nm (excitation) and 480 nm (emission) filters and SkanIt software. For subsequent analysis of the activity measurements Prism 5.0 (GraphPad Software) was used. For Michaelis-Menten analysis of Met-Ala-AMC (MA-AMC) or Met-Pro-AMC (MP-AMC) hydrolysis, 12,5 nM purified recombinant DPP9-short was incubated with various concentrations of MA-AMC or MP-AMC and fluorescence release was measured as described above. Each assay was performed at least three times, each time in triplicates (technical repetitions).

## Peptidase activity assay by liquid chromatography-tandem mass spectrometry (LC/MS/MS)

50 µM of the Syk amino terminus peptide 1–31 (MASSGMADSANHLPFFFGNITREEAEDYLVQ) was incubated alone, in the presence of 130 nM DPP9 wt or its inactive variant DPP9 S730G. To test for inhibition, 10 µM peptide inhibitor (SLRFLYEG) was added. All reactions were performed in TB buffer (20 mM HEPES/KOH, pH 7.3, 110 mM potassium acetate, 2 mM magnesium acetate, 0.5 mM EGTA) supplemented with 0.2% Tween 20. Reactions were stopped after 6 hr by dilution and acidification in aqueous 0.1% formic acid, 2% acetonitrile (1/500, v:v). The resulting samples were analysed on a nanoLC425 nanoflow chromatography system coupled to a TripleToF 5600+ Plus mass spectrometer of QqToF geometry (both SCIEX). In short, 5 µl of sample were pre-concentrated on a self-packed Reversed Phase-C18 precolumn (Reprosil C18-AQ, Pore Size 120 Å, Particle Size 5 µm, 4 cm length, 0.15 cm I.D., Dr. Maisch) and separated on a self-packed Reversed Phase-C18 microcolumn (Reprosil C18-AQ, 120 Å, 3 µm, 15 cm, 0.075 cm) using a 15 min linear gradient (5 to 50% acetonitrile, 0.1% formic acid modifier, flow rate 300 nl/min, column temperature 50°C) followed by a 5 min

high organic cleaning step and a 15 min column re-equilibration. The eluent was introduced to the mass spectrometer using a Nanospray III ion source with Desolvation Chamber Interface (SCIEX) via a commercial Fused Silica tip (FS360-20-10-N, New Objective) at a spray voltage of 2.4 kV, a sheath gas setting of 12 and an interface heater temperature of 150°C. The MS acquisition cycle consisted of a 500 ms TOF MS survey scan that was used for profiling of substrate and product concentrations followed by data-dependent triggering of up to five 100 ms TOF product ion spectra to confirm the identity of detected analytes. Data analysis was performed using Analyst TF 1.7 and PeakView 2.1 softwares (SCIEX). Analyses were performed in triplicates.

## CHX chase assays

$0.5–1 \times 10^6$ DG-75 cells/ml were seeded in 24 wells. 24 hr later cells were treated with CHX (100 µg/ml) and, where stated, in parallel stimulated with 12 µg/ml F(ab')$_2$ fragment goat anti-human IgG + IgM (H+L) (Dianova) in the presence of either 10 µM 1G244 or 100 µM MG132 or DMSO as MOCK control. Cells were harvested at different time points, counted and shock-frozen in liquid N$_2$. Cell pellets were resuspended in the respective amount of sample buffer according to cell number. To analyse the stability of phosphorylated Syk cells were incubated in serum-free RPMI for 20 min before they were seeded in 24 wells ($0.8 \times 10^6$ cells per well), treated with CHX and in parallel stimulated in the presence or absence of 10 µM 1G244 as described above. Cells were harvested at different time points in 250 µl sample buffer. For assays with HeLa wt and DPP9-kd cells $1.4 \times 10^5$ cells were seeded in 24 wells and transfected with various Syk-FLAG constructs together with a GFP plasmid on the next day. 48 hr after transfection, cells were treated with CHX and harvested at the respective time in sample buffer.

## Immunoprecipitations

$3 \cdot 10^7$ DG-75 cells were washed with PBS, resuspended in serum-free RPMI in the presence or absence of 1G244 (10 µM). Unless otherwise stated, cells were incubated at 37°C with the inhibitor for 20 min. Cells were resuspended in cold lysis buffer (20 mM Tris/HCl (pH 7.5), 150 mM NaCl, 0.5 mM EDTA, 10 mM NaF, 10 µM MoO4, 1 mM Na3VO4, 1% IGEPAL CA-630, PMSF, Aprotinin, Leupeptin), and incubated at 4°C for 1 hr with constant agitation. Cell lysates were centrifuged at 50000 g, at 4°C for 15 min. The supernatant was incubated for 30 min at 4°C with protein A beads to remove precipitating proteins. The pre-cleared supernatant was then subjected to IP with 5 µg/ml mouse anti-pTyr 100 (#9411) for 2 hr at 4°C with constant agitation. This was followed by the addition of 20 µl protein A/G-magnetic beads for 30 min (Thermo Fisher Scientific). Following careful washing, proteins were eluted with reducing 2 × sample buffer at 65°C for 5 min.

## Ca$^{2+}$ measurements

$10^6$ DG-75 or Ramos B cells were loaded in 700 µl RPMI containing 5% FCS, 1 µM Indo1-AM (Molecular Probes), and 0.015% Pluronic F127 (Molecular Probes) at 30°C for 25 min. Subsequently, the cell suspension was diluted two-fold with RPMI 10% FCS and incubated for 10 min at 37°C. Cells were washed, treated with 10 µM 1G244 (or mock-treated) and prepared for measurements as described earlier (*Stork et al., 2007*). For BCR-induced Ca$^{2+}$ mobilization, cells were treated with 10 µg/ml F(ab)$_2$ goat-anti-human IgM (Jackson Immunoresearch). Changes in the ratio of fluorescence intensities at 405 nm and 510 nm were monitored on a LSRII flow cytometer (Becton Dickinson) and analysed with FlowJo (TriStar).

## Acknowledgements

We are very grateful to Ernst Jarosch, Jürgen Wienands and Frauke Melchior for critically reading this manuscript. The authors also thank Magdalena Schacherl for helpful discussions. RGF is very thankful to Frauke Melchior for her support in the initial step of this project, and to Blanche Schwappach for her generous continues support.

# Additional information

### Funding

| Funder | Grant reference number | Author |
|---|---|---|
| Deutsche Forschungsge-meinschaft | 2234/1-2 | Ruth Geiss-Friedlander |
| Heidenreich von Siebold-Programm, Universitaetsmedizin Goettingen | | Ruth Geiss-Friedlander |

The funders had no role in study design, data collection and interpretation, or the decision to submit the work for publication.

### Author contributions

DJ-S, MS-G, EP, ME, MK, CL, FN, Acquisition of data, Analysis and interpretation of data, Drafting or revising the article; UM, Acquisition of data, Drafting or revising the article; HU, Conception and design, Drafting or revising the article; RG-F, Conception and design, Acquisition of data, Analysis and interpretation of data, Drafting or revising the article

### Author ORCIDs

Christof Lenz, http://orcid.org/0000-0002-0946-8166
Ruth Geiss-Friedlander, http://orcid.org/0000-0002-1720-3440

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
