## [Decision Letter]

Thank you for submitting your article "DPP9 is a novel component of the N-end rule pathway targeting the Tyrosine Kinase Syk" for consideration by *eLife*. Your article has been favorably evaluated by Vivek Malhotra (Senior Editor) and three reviewers, one of whom is a member of our Board of Reviewing Editors. The following individual involved in review of your submission has agreed to reveal their identity: Guy Salvesen (Reviewer #3).

The reviewers have discussed the reviews with one another and the Reviewing Editor has drafted this decision to help you prepare a revised submission.

This manuscript focuses on a novel function for DPP9 – as a regulator of N-end rule degradation. The search for functioning protease substrates – including DPP9 – is proceeding apace, but as the authors point out the identification of physiologically relevant substrates and partners is hampered by current limitations in technology. The authors have compiled several innovative strategies to explore a physiologic function for DPP9. They find that DPP9 regulates the levels on Syk in a FLNA-dependent manner, and the overall thrust of the paper supports a role for DPP9 in N-end rule degradation.

The reviewers all agreed that this paper was interesting and appropriate, in principle, for *eLife*, subject to addressing a few issues.

1) The link with filamin A is addressed only in the first part of the paper: is there any evidence of similar FLNA involvement in B cells?

2) The demonstration that DPP9 cleaves Syk directly is dependent on a small peptide for the Syk N-terminus. This is central to the paper and it would be reassuring to know that full length Syk is also an efficient substrate.

3) The link between DPP9's activity on Syk and the biological consequences is not very strong. What are the consequences on B cell fate and function of the calcium defects observed? Are other signalling pathways that involve Syk affected (e.g. Ras/MAPK or PKC/NFkB)? Does DPP9 inactivation change cell proliferation or other behaviour in B cells or Syk expressing cancer cells?

4) Much weight is put on the IG244 inhibitor in functional assays. It would be reassuring to use another inhibitor or – even better – genetic knockdown to insure against misleading off-target effects.

---

## [Author Response]

*The reviewers all agreed that this paper was interesting and appropriate, in principle, for eLife, subject to addressing a few issues.*

*1) The link with filamin A is addressed only in the first part of the paper: is there any evidence of similar FLNA involvement in B cells?*

We addressed this question by performing PLAs in DG-75 B cells. These assays confirm that the interactions between FLNA-DPP9 and FLNA-Syk are indeed conserved in the B cells. The new data are added to Figure 6.

*2) The demonstration that DPP9 cleaves Syk directly is dependent on a small peptide for the Syk N-terminus. This is central to the paper and it would be reassuring to know that full length Syk is also an efficient substrate.*

We agree that the cleavage of Syk by DPP9 is a central point for this paper. As suggested we cloned Syk for expression of the full-length protein in *E. coli,* as well as a shorter fragment expressing the two SH2 domains in the N-terminus of Syk fused to a C-terminal GST protein. The purified proteins were incubated with DPP9, control reactions contained Syk alone, or Syk which was incubated with an inactive DPP9 variant or in the presence of an inhibitor. Mass spectrometry analysis of these reactions revealed that the amino terminus of Syk is processed in bacteria,which remove the initiator methionine (more than 90% efficiency). Consequently, the recombinant Syk which we purified from bacteria is unfit for in vitro cleavage assays.

In parallel we addressed this point, by exchanging the Ala in position 2 of Syk N-terminus to Asp (Syk A2D), thus destroying the DPP9 cleavage site in Syk. Pulse-chase analysis of this Syk variant show that abolishing the DPP9 cleavage site strongly stabilizes Syk, supporting the notion that Syk is indeed cleaved by DPP9 in the context of the full protein. These data are added to Figure 5, and are in agreement with results that we have already included in the manuscript: (1) PLA in Figure 3 showing that the interactions between endogenous DPP9 and Syk are reduced in cells that were treated with a DPP9 inhibitor. In this case either a competitive inhibitor (1G244) or an allosteric inhibitor (SLRFLYEG). (2) Functional assays showing that silencing or inhibition of DPP9 clearly stabilize Syk, both in HeLa cells (transfected Syk) and in B cells (endogenous Syk).

Moreover, when examining the published solved structures of Syk, specifically at the amino terminus we observed that the best coverage of Syk N-terminus starts from residue 8, which is followed by an unstructured region. We presume that the lack of solved structures, which include residues 1-7 is due to a high flexibility of this region. The fact that the amino terminus appears to be unstructured, suggests that it is highly flexible, which would strongly support the accessibility of the N-terminus into the active site of DPP9.

To point this out we included a short sentence regarding the N terminus of Syk into the manuscript:

“Based on published structures, the amino terminus of Syk includes an unstructured region followed by an α-helix (residues 22-31) (Gradler et al., 2013; Fütterer et al., 1998). Of note, amino acids 1-7 are not resolved in any determined structure of Syk, suggesting that this region is flexible and thus accessible for interactions.”

Hence the unstructured nature of Syk N-terminus supports the conclusions from the mass spec analysis using small peptides.

Finally, as processing of the full Syk protein could not be reliably tested, we now analyzed whether DPP9 can process a longer synthetic peptide covering the N-terminal residues 1-31 of Syk. This is the peptide we used for the SPR assays. We now show that indeed DPP9 can efficiently cleave a larger Syk peptide, which most possibly mimics the N-terminal structure of the protein, with an unfolded region and an α-helix. These new data replace the previous mass spec analysis in Figure 3, which previously analyzed the cleavage of a shorter peptide (1-15).

Taken together, the mass spec analysis with a long Syk peptide (1-31, Figure 3), the SPR with longSyk peptides (1-31, 3-31), the PLA data (Figure 2: endogenous proteins, use of different inhibitors), the Pulse chases in the DPP9 stably silenced cells and the different variants of Syk, functionally show that DPP9 does cleave Syk in the context of the full length protein and that this cleavage is biologically relevant.

*3) The link between DPP9's activity on Syk and the biological consequences is not very strong. What are the consequences on B cell fate and function of the calcium defects observed? Are other signalling pathways that involve Syk affected (e.g. Ras/MAPK or PKC/NFkB)? Does DPP9 inactivation change cell proliferation or other behaviour in B cells or Syk expressing cancer cells?*

To address the question regarding the biological consequences of DPP9 activity on B cell fate, we tested the effect of DPP9 inhibition on Syk signaling.

Intriguingly we found that DPP9 appears to stabilize preferentially active, phosphorylated Syk, both in stimulated and non-stimulated cells. The calcium defects that are observed upon inhibition of DPP9 in non-stimulated cells appear to result from a mal-regulation of ‘tonic’ Syk signaling in resting cells, as seen by higher levels of phosphorylated Syk. We now complete the picture by showing that also the levels of phosphorylated PLCγ2 are higher upon DPP9 inhibition in resting cells, either with a competitive or with a non-competitive allosteric DPP9 inhibitor. The consequence is a lower sensitivity of the cells to BCR stimulation. This is evident by lower calcium release in stimulated cells (DG-75 and Ramos cells). Moreover, we analyzed more distal BCR events and show that inhibition of DPP9 leads to augmented ‘tonic’ and more transient BCR-induced ERK1/2 signaling, as measured by reduced phosphorylation of ERK1/2.

These new data (Figure 7) further confirm our conclusion that DPP9 is a novel negative regulator of Syk, which keeps the levels of phosphorylated Syk low in unstimulated cells, and consequently attenuates the Syk mediated signal transduction after BCR engagement.

*4) Much weight is put on the IG244 inhibitor in functional assays. It would be reassuring to use another inhibitor or – even better – genetic knockdown to insure against misleading off-target effects.*

We addressed this question from two aspects. Since in HeLa cells we showed specificity using the DPP9 silenced cells, we now concentrated on DG-75 cells. Of note, 1G244 is considered the most specific competitive inhibitor of DPP8 and DPP9. All other available competitive inhibitors target DPPIV as well, due to the high conservation of the active site.

First, we performed PLAs in DG-75 cells and tested whether Syk interacts with 2 homologs of DPP9: DPP8 (60% homology to DPP9, localized to the cytosol and nucleus) and DPPIV (26% homology to DPP9, localized to the plasma membrane). These assays show that signals for DPP8-Syk or DPPIV-Syk are not above background levels of control PLAs, whereas the interaction between DPP9 and Syk was significantly higher than background. These data are inserted into Figure 6.

Second, we performed cellular assays with the allosteric inhibitor SLRFLYEG which enters DG-75 cells, when complexed with pep-1. Figure 7 shows that like 1G244 also SLRFLYEG treatment results in increased steady state levels of phosphorylated PLCγ2 in B cells.